# STUFFED MAMBA: STATE EXPLOSION AND STATE CAPACITY OF RNN-BASED LONG-CONTEXT MODELING

## ABSTRACT

One essential advantage of recurrent neural networks (RNNs) over transformer-based language models is their linear computational complexity concerning the sequence length, which makes them much faster in handling long sequences during inference. However, most publicly available RNNs (e.g., Mamba and RWKV) are trained on sequences with less than 10K tokens, and their effectiveness in longer contexts remains largely unsatisfying so far. In this paper, we study the cause of the inability to process long context for RNNs and suggest critical mitigations. First, we investigate *state explosion* (SE) in Mamba-2 when processing long sequences, a phenomenon where some channels of the state exhibit exploding values that cause severe performance degradation. With controlled experiments, we discover that the model fails to forget the earlier tokens when there is more information than it can remember. We attribute this to overfitting due to the recurrent state being overparameterized for the training length, thereby establishing a relationship between SE and the capacity of the state. To support this hypothesis, we make an important empirical observation: for any given state size, there exists a training length threshold such that SE is exhibited if and only if the training length is greater than this threshold. Empirically searching for this threshold for different state sizes reveals that it is a linear function of the state size. We also search for the maximum context length at which the model can recall contextual information and find that this context length scales exponentially to the state size. Based on this, we empirically train a Mamba-2 370M with near-perfect passkey retrieval accuracy on 256K context length. This suggests a promising future for RNN-based long-context modeling. Code and model checkpoints will be publicly released.

## 1 INTRODUCTION

Recent transformer-based language models (Vaswani et al., 2017; Brown et al., 2020; Touvron et al., 2023; Achiam et al., 2023; Dubey et al., 2024) have demonstrated impressive capabilities in reasoning over long sequences with thousands and even millions of tokens (Team, 2024a;b; GLM et al., 2024). However, they rely on the attention mechanism that scales quadratically regarding the sequence length, making them extremely costly for inference over long sequences. In contrast, recurrent neural networks (RNNs) (Bengio et al., 1994) have a contextual memory with constant state size. Thus, during inference, their per-token computational and memory complexity scales linearly with the sequence length, making them much more efficient in processing long sequences.

Despite the great efficiency of RNNs in processing long contexts, their long-context performances are far from satisfying. Most recent state-of-the-art (SOTA) RNN-based language models (hereafter referred to as *RNNs* for simplicity), such as Mamba-1 (Gu and Dao, 2023), Mamba-2 (Dao and Gu, 2024), the RWKV series (Peng et al., 2023a; 2024), and GLA (Yang et al., 2023) are trained on sequences with less than 10K tokens. Existing works have shown that Mamba-1 and RWKV-4 suffer from severe performance drops when the context length exceeds their training length (Ben-Kish et al., 2024; Zhang et al., 2024a; Waleffe et al., 2024).

In this paper, we study the problem that what causes the current RNNs' inability to handle long contexts and what are the possible solutions for supporting long contexts. When applying RNNs to longer contexts, we observe two critical problems. (1) RNNs are unable to generalize along the sequence length. They exhibit abnormal behavior when the context length exceeds the training length,

resulting in poor long-context performance. (2) Since their memory size is constant, although they can process infinitely long inputs, there is an upper bound to the amount of information the state can represent. Therefore, there is an upper bound of the contextual memory capacity (the maximum number of tokens that can be remembered), and tokens beyond that limit will be forgotten.

Then we dive deeper into the formation of the above problems. We first attribute the cause of the length generalization failure of SOTA RNNs to a phenomenon we call *state explosion* (SE). It describes the exploding channels in the recurrent state that cause performance degradation as the model consumes too many tokens. We inspect the memory state distribution over time and discover that the explosion is largely contributed by a few dominant outlier channels. These outliers cause vanishing values in other channels when the output hidden representation is normalized. By analyzing various components of the state update rule, we show that SE is caused by the inability to forget the earliest tokens (by decaying the state with a smaller multiplier) when there is more information than it can remember. To support this finding, we empirically show that it is possible to mitigate SE without training by forcing the model to forget contextual information by (1) reducing the memory retention and insertion strength and (2) reformulating the recurrence into an equivalent sliding window state (which makes it forget everything outside of the window). These methods allow Mamba-2 to consume more than 64K tokens without SE.

Additionally, we present the *state overparameterization hypothesis* as a possible explanation for why the model fails to learn a robust forgetting mechanism. This hypothesis tells us that one can avoid SE by training on sufficiently long sequences that exceed the model's state capacity, which forces the model to learn how to forget excessive information to prevent SE. We empirically validate this hypothesis by discovering that for any state size, there is a threshold for training length beyond which the model will not exhibit SE. This insight allows us to establish a relationship between state capacity and state size. Then, by training Mamba-2 models of different sizes, we establish that the empirical capacity is a linear function of the state size. Furthermore, we conduct the same experiments on the widely used passkey retrieval task (Mohtashami and Jaggi, 2023), and show that the length where Mamba-2 has near-perfect passkey retrieval accuracy is an exponential function of the state size. The experiment results in a Mamba-2 370M model that can achieve near-perfect passkey retrieval accuracy on 256K context length, significantly outperforming transformer-based models of the same size in both retrieval accuracy and length generalizability. Our results show that the commonly used training lengths for RNN-based models may be suboptimal and that RNN-based long-context modeling has promising potential.

The main findings of this paper can be summarized as follows.

**State explosion**: In Mamba-2, there exists a small number of channels with exploding values that cause severe performance degradation on sequences longer than the training length. (Section 4)

**State overparameterization hypothesis**: By analyzing the hidden representations, we find that state explosion occurs because the model fails to forget when the state is being overflowed with excessive information. We attribute this to state overparameterization, which establishes the connection between SE and state capacity. (Section 4.3)

**Training length threshold**: In alignment with the state overparameterization hypothesis, we empirically discover that for any state size, there exists a training length threshold where SE is exhibited if and only if the training length is below that threshold. (Section 5)

**Relationship between state capacity and state size**: Utilizing the previous finding, we can empirically search for the relationship between the training length threshold and state size, and discover that it is linear. We also empirically search for state capacity on passkey retrieval, and find that it scaled exponentially with the state size. (Section 7.1)

## 2 RELATED WORKS

**RNN-Based Language Models**   There is a recent surge of interest in RNN-based language models, because, contrary to transformer-based ones, their per-token inference cost does not increase with the sequence length. Linear Attention (Katharopoulos et al., 2020) replaces the softmax attention in transformer-based models with kernel-based approximations that have equivalent recurrent formulations. Some notable recent RNNs include the RWKV series (Peng et al., 2023a; 2024), the Mamba

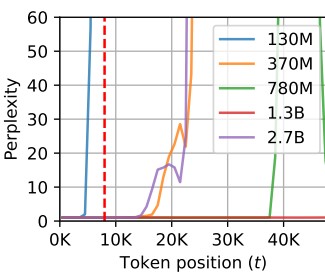 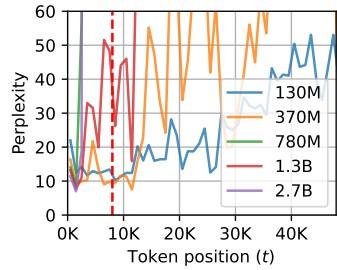 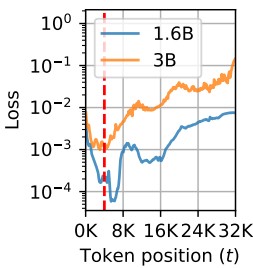

(a) Mamba-2 Series on the "newlines" prompt (only "\n" characters).

(b) Mamba-2 series on long documents from RedPajama.

(c) RWKV-6 1.6B and 3B on the "newlines" prompt.

Figure 1: The perplexity of Mamba-2 and RWKV-6 as a function of token position on real and synthetic data. The red dotted line represents the models' training lengths. They all fail to extrapolate.

series (Gu and Dao, 2023; Dao and Gu, 2024), Gated Linear Attention (Yang et al., 2023), among others (Zhang et al., 2024b; Yang et al., 2024; De et al., 2024; Arora et al., 2024; Orvieto et al., 2023; Sun et al., 2023). These models have shown strong capabilities in many language processing tasks, sometimes outperforming transformer-based models. However, as we will empirically show, some of these models fail to extrapolate much beyond their training length.

Some transformer-based models have adopted sliding window attention (Beltagy et al., 2020; Jiang et al., 2023), which essentially turns them into RNNs. However, these models have been shown to perform poorly in long-context tasks and fail to extrapolate to very long contexts (Zhang et al., 2024a).

**Length Generalization**    Most SOTA language models in the last few years have been based on the transformer (Vaswani et al., 2017) architecture. These models, when using certain variants of position encoding, can process arbitrarily long sequences. However, they exhibit severe performance drops on tokens beyond the training length (Zhao et al., 2024). To alleviate this shortcoming, many works have focused on modifying positional encoding (Peng et al., 2023b; Zhu et al., 2023; Ding et al., 2024; Jin et al., 2024), some achieving training-free length generalization to certain extents. Similarly in spirit, this study also explores some post-training modifications for enhancing RNN-based models' length generalization capabilities.

**Length Generalization of Mamba**    Some prior works investigated the performance of Mamba as a function of context length (Park et al., 2024; Wen et al., 2024). Jelassi et al. (2024) empirically showed a sharp drop in performance beyond the training length for Mamba and LSTM on a copying task and also showed that Mamba struggles to copy from context unless its state size grows linearly with the context length. Arora et al. (2023) discussed the capacity and empirical ability to perform associative recall of transformer and some recurrent language models. In contrast, our paper explores the empirical relationship between state capacity and length generalization failure.

Some concurrent works have explored extending Mamba's context length by controlling the discretization term ($\Delta_t$ in Eq. 3) (Ben-Kish et al., 2024), such as dividing it by a constant to make it smaller[1]. This essentially makes the memory decay factor ($\alpha_t$ in Eq. 5) closer to 1, which makes the state retain more contextual information. However, it also unnecessarily diminishes the inserted information on all tokens. Consequently, although it can mitigate SE, it results in poor performance in the passkey retrieval task (details in Appendix C).

## 3   PRELIMINARY

Most experiments in this study focus on Mamba-2 (Dao and Gu, 2024) because it has shown strong capabilities on several tasks and has publicly available checkpoints of multiple sizes, allowing us to

---

[1] https://github.com/jzhang38/LongMamba

explore the relationship between state sizes and length limits. Moreover, it is more widely studied than other RNNs, making it easier to use existing works as a reference.

**Mamba-2** The Mamba-2 architecture consists of $L$ layers, each consisting of $H$ heads computed in parallel. The layer's output is the sum of the heads' outputs. Each head can be formulated as follows.

$$y_t = \text{Norm}(o_t \odot u_t W_{\text{gate}})W_o \in \mathbb{R}^d \tag{1}$$

$$o_t = C_t h_t + D \odot x_t \in \mathbb{R}^P \tag{2}$$

$$h_t = h_{t-1}\alpha_t + \overline{B}_t x_t \in \mathbb{R}^{N \times P} \tag{3}$$

where $t$ denotes the current time step, $u_t, y_t \in \mathbb{R}^d$ are the input and output hidden representations of the $t$-th token, $\text{Norm}(\cdot)$ denotes the RMS normalization (Zhang and Sennrich, 2019), $D \in \mathbb{R}^P$, $W_{\text{gate}} \in \mathbb{R}^{d \times P}$, and $W_o \in \mathbb{R}^{P \times d}$ are trainable parameters, $\odot$ denotes element-wise product, and $d, N, P$ are hyperparameters, denoting the hidden dimensionality, state dimension, and head dimension, respectively. The other variables are parameterized as follows:

$$\overline{B}_t = \Delta_t B_t \in \mathbb{R}^{N \times 1} \tag{4}$$

$$\alpha_t = \exp(-\Delta_t \exp(A)) \in \mathbb{R} \tag{5}$$

$$C_t = \sigma(\text{Conv}(u_t W_C)) \in \mathbb{R}^{1 \times N} \tag{6}$$

$$B_t = \sigma(\text{Conv}(u_t W_B)) \in \mathbb{R}^{N \times 1} \tag{7}$$

$$x_t = \sigma(\text{Conv}(u_t W_x)) \in \mathbb{R}^{1 \times P} \tag{8}$$

$$\Delta_t = \text{Softplus}(u_t W_\Delta + b_\Delta) \in \mathbb{R} \tag{9}$$

where $\left(W_C, W_B \in \mathbb{R}^{d \times N}, W_x \in \mathbb{R}^{d \times P}, W_\Delta \in \mathbb{R}^{d \times 1}, b_\Delta, A \in \mathbb{R}\right)$ are trainable model parameters. $\text{Conv}(\cdot)$ denotes a channel-wise one-dimensional convolutional layer (see Appendix A.2). $\sigma$ denotes the SiLU function (Elfwing et al., 2017). Importantly, $h_t$ is called the *hidden state* or *recurrent state*, which is contextual memory that stores information from all tokens up to $t$. $\alpha_t$ is the decay multiplier that controls the strength of memory decay (i.e., forgetting). Appendix A presents more details.

It is worth mentioning that this update rule can be seen as a variant of Gated Linear Attention (Yang et al., 2023) and is similar to many existing RNNs (e.g., RWKV (Peng et al., 2024) and RetNet (Sun et al., 2023)). Thus, some conclusions/insights may apply to other architectures. We leave such exhaustive ablation studies for future work.

## 4 STATE EXPLOSION

We first evaluate Mamba-2 and RWKV-6 on language modeling and passkey retrieval and find that they fail to extrapolate beyond their training lengths. Then, we analyze the state's distribution on different context lengths and find that a few outlier channels exhibit exploding values. Based on this finding, we coin the term *state explosion*, which describes the phenomenon where the recurrent state exhibit exploding values, causing performance degradation.

By analyzing the components of the state's update rule, we discover that SE is caused by the failure to forget past information when the context is too long. We then propose a hypothesis that explains SE from the perspective of overparameterization.

### 4.1 LENGTH GENERALIZATION FAILURE

**Language Modeling** Figure 1 shows the language modeling loss of Mamba-2 and RWKV-6 beyond their training lengths. For controllability and to synthesize prompts of arbitrary lengths, this loss is computed on a prompt consisting of only the "\n" characters, which we refer to as the "newlines" prompt. However, we emphasize that the same observation also exists when processing texts from the pre-training corpus. The result shows that both RNNs suffer great performance degradation when the context length is much longer than their training lengths.

**Passkey Retrieval Evaluation** Language modeling may not reflect downstream capabilities, thus, we evaluate several strong RNNs on the passkey retrieval task (Mohtashami and Jaggi, 2023; Zhang

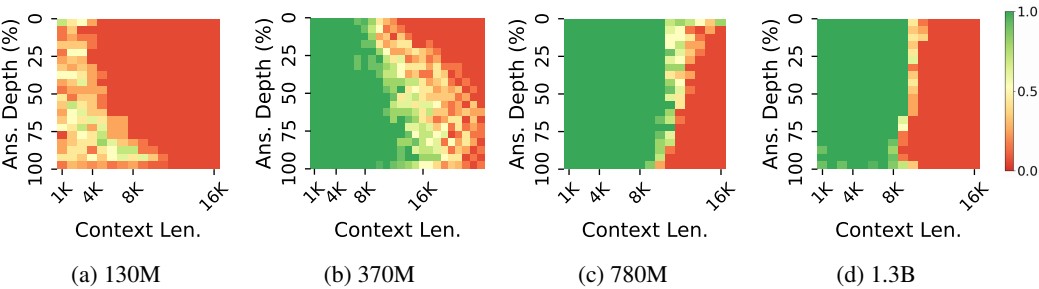

(a) 130M       (b) 370M       (c) 780M       (d) 1.3B

Figure 2: The performance of Mamba-2 official checkpoints on the passkey retrieval task. "Ans. Depth" refers to the position of the passkey relative to the context length, 0% means that it is located at the start of the context, 100% means that it is at the end.

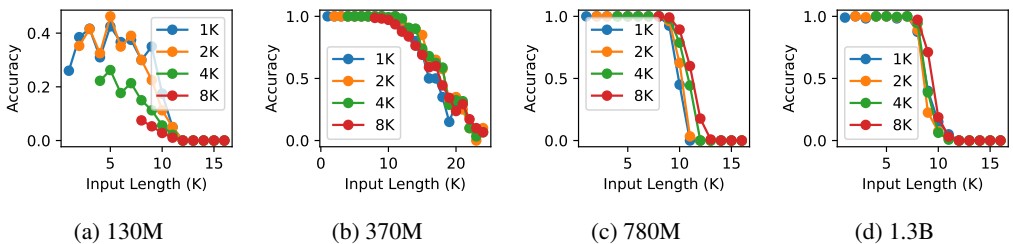

(a) 130M       (b) 370M       (c) 780M       (d) 1.3B

Figure 3: The accuracy of Mamba-2 official checkpoints on the passkey retrieval task where the answer is in the last $n$ tokens, with $n$=1K, 2K, 4K, and 8K.

et al., 2024a), a simple synthetic task where a model is prompted to recall a 5-digit *passkey* from a lengthy context. The hyperparameters and results of other RNNs can be found in Appendix B and D. The results for Mamba-2 are reported in Figure 2. We find that Mamba-2 models fail to generalize to sequences longer than the training length. For instance, Mamba-2, trained on 8K context windows, has near-perfect retrieval accuracy within 8K contexts (except for the smaller 130M checkpoint), but poor or even zero accuracy on sequences longer than 16K, regardless of model sizes.

This behavior is unexpected because the update rule (Eq. 3) has a stable exponential memory decay (it converges to a constant value if the variables are fixed). Therefore, we expect RNNs of such form to have a good retrieval accuracy on the last $k$ tokens, and tokens earlier than that are forgotten. This unexpected finding also implies that when processing contexts longer than the training length $T_{\text{train}}$, it is better to keep just the last $T_{\text{train}}$ tokens and discard everything else. However, this is not trivial in an online inference scenario because all token information is compressed into a single state.

## 4.2 OBSERVATION OF STATE EXPLOSION

Since the recurrent state's dimensionality does not change over time, the sharp change of behavior during length generalization must be a result of a change in the state's value. We inspect the statistics of the recurrent states of each layer in Mamba-2 370M[2] and find that the mean and variance of some heads change sharply when the context length exceeds the training length. One example is shown in Figure 4[3]. Appendix G reports the statistics of each layer. The state at $t = 20K$ of one head with exploding variance is shown in Figure 5. From it, we discover that this variance explosion can be largely attributed to a few outlier channels, while most channels are relatively stable.

We emphasize that **SE occurs largely independent of the prompt**, occurring in both pre-training data samples and generated meaningless texts, even for prompts consisting of only whitespace characters

---

[2]We are using this model size as an example, but the same observation is found with any model size.

[3]We emphasize that we are showing only a subset of heads for one layer in this figure to keep it clean. Exploding heads are found in other layers as well.

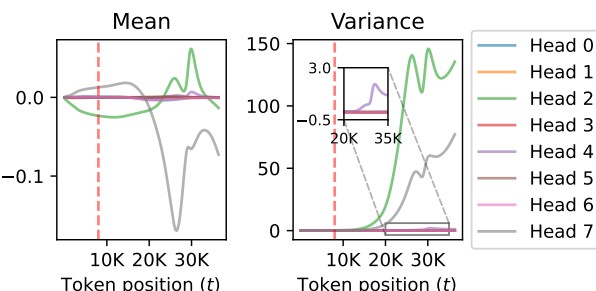 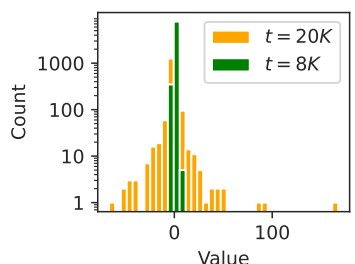

Figure 4: The mean and variance of the first 8 heads in the 38th layer of Mamba-2 370M. It exhibits a clear explosion when $t$ is greater than the training length. The red dotted line indicates the training length.

Figure 5: The distribution of the channels in a collapsing state (head 2 for the 38th layer) at two different time steps.

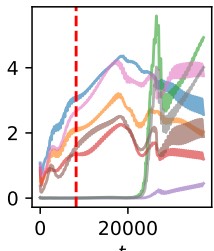 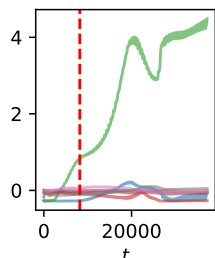 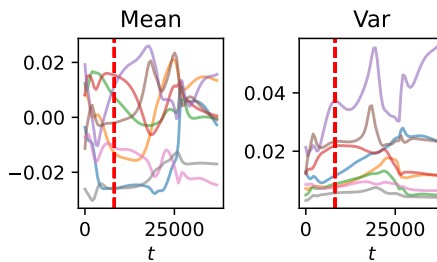

(a) $\Delta_t$. Each curve represents a head.

(b) $B_t$. Each curve represents a channel.

(c) The statistics of $x_t$ over time. Each curve represents a head.

Figure 6: The value of the components in the update rule ($\Delta_t$, $B_t$, and $x_t$) on some heads with SE in the 38th layer in Mamba-2 370M. The red dotted line indicates the training length.

(Figure 1 (a) and (b) shows SE on two different prompts). This means that the information inserted into the state does not cause its explosion.

To further attribute to the specific variables that cause SE, we inspect the values of $\Delta_t$, $B_t$, and $x_t$ on various heads with exploding states. Figure 6 reports one example of the inspection, we can see that $x_t$ is relatively stable compared to the $\Delta_t$ and $B_t$, even though they are all functions of $u_t$. We also notice that $B_t$ explodes earlier than $\Delta_t$ (ignoring the non-exploding heads). Further inspection reveals that the convolutional weights that generate $\Delta_t$ and $B_t$ (Eq. 7 and 9) are noticeably greater in variance than those for $x_t$ (Eq. 8). A more in-depth attribution study is outside the scope of this work.

### 4.3 STATE OVERPARAMETERIZATION HYPOTHESIS

Here, we present a high-level explanation for SE. We argue that **SE arises from state overparameterization** relative to the training length. In other words, the state is excessively large for the training length, allowing the model to achieve strong language modeling performance without learning how to forget when the state is about to overflow.

To support this argument, we formulate the hidden state as a weighted sum of previously inserted information:

$$h_t = \sum_{i=1}^{t} \alpha_{i:t} \overline{B}_i x_i, \quad \alpha_{i:t} = \left( \prod_{j=i}^{t} \alpha_j \right) \in (0, 1) \tag{10}$$

Thus, $\alpha_{i:t}$ describes the memory strength about the $i$-th token at $t$ time step. Figure 7 shows the memory strength of the first token at different time steps, and we find that the exploded heads (heads 2, 4, and 7 in the 38th layer) have a strong inclination toward retaining all information within the

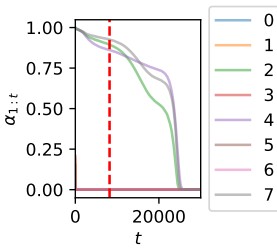

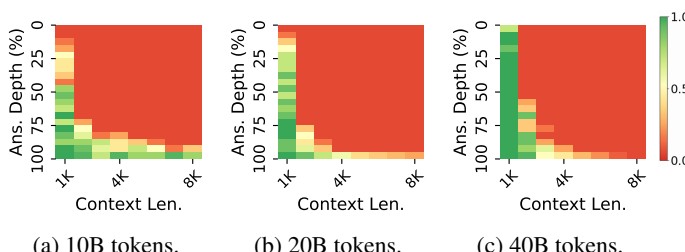

(a) 10B tokens.    (b) 20B tokens.    (c) 40B tokens.

Figure 7: The cumulative decay of the first token ($\alpha_{1:t}$) over time. Each line represents a head. The statistics are collected from the 38th layer of Mamba-2 370M.

Figure 8: The passkey retrieval results of intermediate checkpoints during the pre-training of Mamba-2 370M on 512 sequence length. SE only occurs in the model beyond a certain amount of training data.

training length, with a memory strength of over 0.8 at $t$=8K. This implies that the model has not learned to forget information (by producing a smaller decay $\alpha_j$), but it still has decent language modeling capabilities because the state has the capacity to store all information in 8K tokens.

Furthermore, we pre-train Mamba-2 from scratch with $T_{\text{train}} = 512$ and evaluate the intermediate checkpoints on passkey retrieval, as reported in Figure 8. It shows that SE is only exhibited by checkpoints beyond a certain amount of training, which coincides with behaviors of overfitting—a result of overparameterization. One can also notice that the overfitted checkpoint outperforms earlier checkpoints on shorter sequences, which further strengthens the hypothesis that the model converges to less forgetting.

Finally, in Section 7.1, we empirically find that for any given state size, there exists a length threshold $T_*$ where SE is exhibited if and only if $T_{\text{train}} < T_*$.

### 4.4 HOW TO MITIGATE STATE EXPLOSION?

To further support the state overparameterization hypothesis in the previous section, we demonstrate that it is possible to mitigate SE by simply forcing the model to forget more and/or remember less information. This can be achieved by training on sufficiently long sequences or by training-free modifications to the update rule. We emphasize that the training-free methods serve as demonstrations of the link between forgetting and SE, and we recommend training on longer sequences in practice.

#### 4.4.1 TRAINING ON LONGER SEQUENCES

We employ two techniques to improve effectiveness and efficiency when training on longer sequences.

**Data Engineering**    To ensure that the data contains as much long-term structure as possible, we filter out sequences with less than 4K tokens. Buckman and Gelada (2024) have shown that this is critical for training effective long-context models. Although we train on sequences longer than 4K tokens, we do not use a higher length threshold because the above threshold already removes about 97.6% of the data in the original corpus. To train on longer sequences, we simply concatenate sequences and delimit them with a special EOS (End-of-Sequence) token.

**Truncated Backpropagation Through Time**    In the vanilla Mamba-2, the states are initialized to zeros for each data sample. Instead, we initialize the states as the final state of the previous sequence. This is equivalent to concatenating multiple sequences, but stopping the backpropagation of gradients at certain intervals. This technique has been shown to help extend the context length of RNNs (Yang et al., 2023) and alleviate the memory cost of caching activations for computing gradients. Based on Yang et al. (2023) and our preliminary tests, we use concatenate 12 sequences with this technique by default.

### 4.4.2 TRAINING-FREE MITIGATION METHODS

Here, we propose some tricks for mitigating SE by inducing more forgetting without training. Since the performance of these tricks is not the focus of this paper, we report their evaluation result in Appendix I. The main takeaway from these methods is that SE can be prevented by simply inducing more forgetting by modifying the update rule without additional training.

**Reduced Retention and Insertion Strength (RRI)**   This method assumes that $\alpha_t$ and $B_t$ control the memory retention and insertion strength, respectively. We scale them with a multiplier smaller than 1. The actual value is chosen by validation on the "newlines" prompt. Existing works (**?**) have experimented with modifying $\Delta_t$, but it controls both the insertion and decay strength, making it hard to analyze and control.

**Sliding Window**   We can utilize the fact that the state $h_t$ can be written as a weighted sum (Eq. 10) to simulate a sliding window mechanism without re-processing from the start of the window at every step. Let $w \in \mathbb{N}$ denote the window size and $h_t^{(r)} \in \mathbb{R}^{N \times P}$ denote the hidden state when applying the model on the last $w$ tokens at time step $t$. We can then compute $h_t^{(r)}$ exactly as the difference between two states:

$$h_t^{(r)} = \sum_{i=t-r+1}^{t} \alpha_{i:t}\overline{B}_i x_i = \sum_{i=1}^{t} \alpha_{i:t}\overline{B}_i x_i - \alpha_{t-r+1:t}\sum_{i=1}^{t-r} \alpha_{i:t-r}\overline{B}_i x_i = h_t - \alpha_{t-r+1:t}h_{t-r} \quad (11)$$

During streaming generation, we only have to maintain $(h_{t-1}, h_{t-r}, \alpha_{t-r+1:t})^4$, and advance each of them in parallel. However, directly computing $\alpha_{t:t-r}$ may suffer from instability due to floating-point imprecision. Therefore, we maintain $\Delta_{t-r:t} = \sum_{i=t-r}^{t} \Delta_t$ instead, and re-compute $\alpha_{t-r:t} = \exp\left(-\Delta_{t-r:t}\exp(A)\right)$ at every step, which incurs minimal computational cost.

This method can be applied to all RNNs that can be written as a weighted sum, which includes RWKV 5 and 6, RetNet, GLA, etc. It doubles the computation and memory cost for generation, but we believe that it is an acceptable trade-off because RNNs have a very low generation cost compared to transformer-based models and the context processing cost is unchanged.

## 5 STATE CAPACITY

Based on the discussion and hypothesis in Section 4.3, we can avoid SE by training on sufficiently long sequences such that the amount of contextual information exceeds the capacity of the state, thereby forcing the model to learn how to forget. This hypothesis implies the following law:

> Let $P_S$ and $T_{\text{train}}$ denote the recurrent state size and training length, respectively, there exists a training length threshold $T_*(P_S)$ such that SE is exhibited if and only if $T_{\text{train}} < T_*$.

In Section 7.1, we empirically validated this by sweeping different training lengths for different state sizes, and checking whether the model exhibits SE. Hence, $T_*$ can be viewed as an indirect measurement of state capacity because it tells us when the amount of contextual information exceeds the capacity. Here, it is important to disambiguate two notions of state capacity: the theoretical capacity of the state and the empirically observed capacity on a given task. In this paper, we refer to the latter, which is an empirical description of the amount of context information a state can store without forgetting.

We conduct the training as described in Section 4.4.1. We train multiple Mamba-2 with different state sizes and training lengths to find the relationship between $T_*$ and $P_S$. To determine whether a state has exploded, we feed the "newlines" prompt with 1M tokens to the model, and define explosion as the point where perplexity is more than 2x the maximum perplexity within $T_{\text{train}}$ tokens.

---

[4]We also have to cache the last $r$ token IDs, but their size is negligible compared to $h_{t-1}$ and $h_{t-r}$.

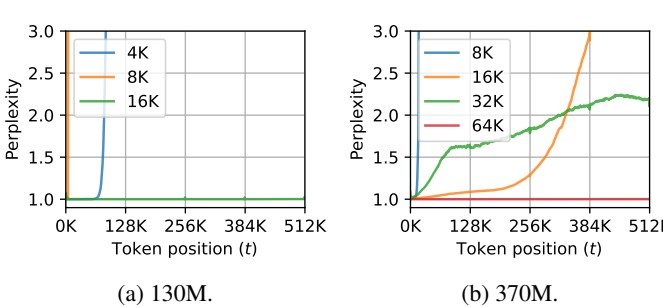 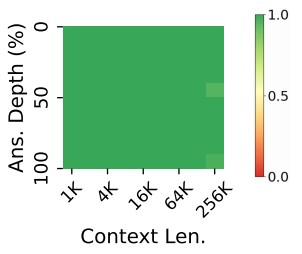

(a) 130M.                    (b) 370M.

Figure 9: The perplexity as a function of token position for Mamba-2 on the "newlines" prompt with different training lengths.

Figure 10: Passkey retrieval performance of Mamba-2 370M. The x-axis is in the log scale.

## 5.1 State Capacity in Passkey Retrieval

It is worth highlighting that the model does not necessarily fail to recall information beyond the last $T_*$ tokens, especially when there is a clear distinction between the target information and other contextual information. Therefore, we also search for the state capacity on the passkey retrieval task, which we regard as an evaluation of the minimum ability to accurately recall a short sequence from a lengthy context (the task is very easy compared to other long-context tasks). Similar to the previous section, we train with different lengths for different state sizes and identify the maximum context length where the model has an accuracy over 95%, which we regard as the *state capacity in passkey retrieval*, denoted with $C_{\text{passkey}}$. In this task, the noisy context is repetitive, thus, the amount of contextual information is largely independent of the context length, therefore, the capacity should grow roughly exponentially with the state size.[5]

## 6 Experiments

We briefly describe the experimental details for training with longer sequences. See Appendix F for more comprehensive experimental details.

**Data**   We start from RedPajama-V2 (Computer, 2023), an open dataset with 30T tokens extracted from CommonCrawl[6], and we perform deduplication to ensure data quality. During evaluation, we sample documents longer than 16K tokens and concatenate them if it is not long enough.

**Models**   We experiment with six model configurations with different state sizes to find the relationship between state capacity and size. For each of them, we perform an extensive search with training lengths up to 256K tokens. To save cost, we continue pre-train from three official checkpoints of Mamba-2 of size 130M, 370M, and 780M. They were pre-trained with 8K sequences. The other three model configurations (36M, 47M, and 85M) are trained from scratch.

**Hyperparameters**   We use the WSD (Hu et al., 2024) with 10% decay steps. This scheduler is chosen because it is competitive with the commonly used cosine scheduler while allowing simple resumption from intermediate checkpoints, saving large amounts of computational resources. We report the result of the best checkpoint selection by validation on passkey retrieval.

---

[5]If we train Mamba-2 on passkey retrieval data, the model can theoretically handle infinitely long contexts. Here, the model is only trained with the next token prediction objective, which means the model will *not* ignore the irrelevant context, and the ability to retain information for extended time emerges from language modeling.

[6]https://commoncrawl.org/

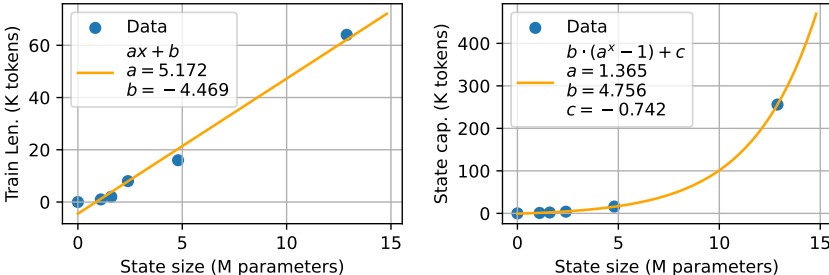

Figure 11: The left figure shows the training length beyond which the model does not exhibit SE (i.e., $T_*$ in Section 5). The right figure shows the state capacity on passkey retrieval (i.e., $C_{\text{passkey}}$) as a function of state size.

## 7 RESULTS

### 7.1 LENGTH GENERALIZATION BY TRAINING ON LONGER SEQUENCES

In Figure 9, we plot the language modeling perplexity as a function of token position for Mamba-2 130M and 370M with different training lengths. We can see that for each model size, there is a training length threshold, beyond which the model has much better length extrapolation, which supports our arguments discussed in Section 4.4.1.

### 7.2 STATE CAPACITY AS A FUNCTION OF STATE SIZE

Figure 11 shows the state capacity of Mamba-2 on language modeling and passkey retrieval. The rightmost data point in both plots corresponds to Mamba-2 370M. We have confirmed that the 780M model (with a state size of 19.3M) also exhibits SE at training lengths below 128K, but do not have enough resources to train the model beyond this length. The results establish a linear relationship $T_* = 5.172 \cdot P_S - 4.469$ between the length $T_{\text{train}} = T_*$ at which SE stops occurring and the state size $P_S$. The $R^2$ value is over 0.999. This indicates that **to train a Mamba-2 with robust length generalization, one should use training lengths are grow linearly with the state size.**

The second plot of Figure 11 shows that the capacity of Mamba-2 on passkey retrieval is exponential concerning the state size, the function is $C_{\text{passkey}} = 4.756 \cdot (1.365^{P_S} - 1) - 0.742$, with an $R^2$ value over 0.999. This is because the amount of information in the context does not increase with its length. In other words, we are storing a constant amount of information while the number of combinations of the state grows exponentially with the number of elements. Figure 10 shows the best checkpoint of Mamba-2 370M on passkey retrieval. The result is very promising because, to the best of our knowledge, no previous models with less than 1B model parameters have near-perfect accuracy at this length in this task.

## 8 CONCLUSION

This paper discovers and presents the first systemic study on *state explosion* (SE), a phenomenon in RNNs that causes length generalization failure. Controlled experiments on Mamba-2 reveals that SE occurs because the model fails to forget when the state is about to overflow, and we conclude that this phenomenon is caused by an overparameterized state with excessive state capacity. Then we show that SE can be mitigated by inducing more forgetting by modifying the update rule or by training on context lengths that exceed the *state capacity*. We discover that for any state size, there exists a training length beyond which SE will not occur. With this insight, we empirically estimate the state capacity of Mamba-2 on language modeling and the passkey retrieval task. With some simple data engineering and state initialization tricks, we achieve much better performance with Mamba-2 on the passkey retrieval task than existing models. Our results indicate that Mamba-2 not only is highly efficient in handling long sequences but also has great performance potential.

## LIMITATIONS

All models studied in this work can be seen as a specific case of *linear attention* models, whose recurrent state is decayed by an element-wise or scalar gate (they can be viewed as variants of Gated Linear Attention (Yang et al., 2023)). We have chosen to study these models because of their strong capabilities, yet, some conclusions may not be directly transferred to other variants of RNNs.

Our continued pre-training approach for extending the context length of RNNs can be rather expensive, some of the models require training with up to 50B tokens, which is 1/6 of their pre-training amount of data. Also, to ensure simplicity, controllability, and generality, we have not used more advanced techniques for training long-context models, such as upsampling longer data samples, better data order or format, using data with more long-distance dependencies, etc.

The passkey retrieval task that we used extensively in this study is very simple. Hence, high accuracy on this task may not reflect the capabilities of the model on real-world long-context tasks, because that requires more advanced capabilities such as high-resolution retrieval, reasoning, state-tracking, etc. The result is nonetheless promising because it indicates that the model can recall the correct information and further capabilities may be achieved by building on the recalled information.

SE is somewhat prompt-dependent. While we found that for models with greatly overparameterized states, SE is highly consistent across different prompts, certain models with less overparameterization may exhibit SE on some prompts while successfully extrapolating indefinitely on others. Attributing SE to specific features of the prompt is a promising future research direction. Moreover, as evident in Figure 1, length generalization failure is less severe in RWKV-6. This is because RWKV-6 has a much smaller state, as shown in Figure 12. Some architectural choices may also affect the occurrence of SE, but we leave such an exploration for future work.

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

## A  MAMBA-2 ARCHITECTURE

For completeness, we give a more detailed formulation of the Mamba-2 architecture here, although we recommend the readers refer to the original paper (Dao and Gu, 2024) or a detailed blog post by the authors[7]. The model accepts a sequence of $T$ token IDs as input $\mathbf{I} = [i_1, \cdots, i_T] \in \mathbb{R}^T$, $i_t \in \{1, 2, \cdots, V\}$, where $V$ denotes the vocabulary size. It performs next token prediction by predicting the probability distribution over the vocabulary as each time step, denoted as $P \in \mathbb{R}^{T \times V}$. The model can be formulated as follows.

$$\mathbf{U}^{(0)} = \mathrm{Embed}_{\mathrm{in}}(\mathbf{I}) \in \mathbb{R}^{T \times d}$$

$$\mathbf{U}^{(l)} = \mathrm{Mamba}^{(l)} \left( \mathrm{Norm} \left[ \mathbf{U}^{(l-1)} \right] \right) \in R^{T \times d}$$

$$\mathbf{P} = \mathrm{Embed}_{\mathrm{out}} \left( \mathrm{Norm} \left[ \mathbf{U}^{(L)} \right] \right) \in \mathbb{R}^{T \times V}$$

where $L$ denotes the number of layers, $l \in \{1, \cdots, L\}$ denotes the layer index, $\mathbf{U}^{(l)} \in \mathbb{R}^{T \times d}$ represents the input of the $l$-th layer, $\mathbf{U}^{(0)}$ represents the input of the first layer. $\mathrm{Mamba}^{(l)}(\cdot)$ denotes the $l$-th Mamba layer, $\mathrm{Embed}_{\mathrm{in}}(\cdot)$ and $\mathrm{Embed}_{\mathrm{out}}(\cdot)$ denote the input and output embedding layers, and $\mathrm{Norm}(\cdot)$ denotes RMS normalization (Zhang and Sennrich, 2019). $d$ denotes the number of dimensions of each token embedding. Similar to many other models, Mamba-2 ties the weight of the input and output embedding layers.

Each Mamba layer consists of $H$ "heads" that are computed in parallel. The result of which is summed together. The $t$-th token ($t \in \{1, \cdots, T\}$) in a head is computed as follows.

$$y_t = \mathrm{Norm}(o_t \odot u_t W_{\mathrm{gate}}) W_o \in \mathbb{R}^d$$

$$o_t = C_t h_t + D \odot x_t \in \mathbb{R}^P$$

$$h_t = h_{t-1} \exp(-\Delta_t \exp(A)) + \Delta_t B_t^T x_t \in \mathbb{R}^{N \times P}$$

$$C_t = \sigma(\mathrm{Conv}(u_t W_C)) \in \mathbb{R}^N$$

$$B_t = \sigma(\mathrm{Conv}(u_t W_B)) \in \mathbb{R}^N$$

$$x_t = \sigma(\mathrm{Conv}(u_t W_x)) \in \mathbb{R}^P$$

$$\Delta_t = \mathrm{Softplus}(u_t W_\Delta + b_\Delta) \in \mathbb{R}$$

$u_t$ denotes the $t$-th input representation. In other words, for the $l$-th layer, we have $\mathbf{U}^{(l)} = \left[ u_1^{(l)}, \cdots, u_T^{(l)} \right]$, $\mathbf{U}^{(l+1)} = \left[ y_1^{(l)}, \cdots, y_T^{(l)} \right]$, and $u_t^{(l+1)} = y_t^{(l)}$. $\mathrm{Conv}(\cdot)$ denotes a channel-wise one-dimensional convolutional layer with a kernel size of 4, and $\sigma$ denotes the SiLU activation function (Elfwing et al., 2017). All vectors are row vectors, and the result of a matrix multiplied by a scalar is the matrix with each element multiplied by that scalar.

$\left( W_{\mathrm{gate}}, W_x \in \mathbb{R}^{d \times P}, W_o \in \mathbb{R}^{P \times d}, W_C, W_B \in \mathbb{R}^{d \times N}, W_\Delta \in \mathbb{R}^{d \times 1}, b_\Delta, A \in \mathbb{R} \right)$ are trainable parameters of the layer, and $P, N$ are hyperparameters. The authors call $P$ the *head dimension* and $N$ the *state size*. In practice, the weights of $W_B, W_C$ are shared among different heads.

### A.1  STATE SIZE

The authors of Mamba-2 always set $P = 64$, $N = 128$, and $H = 2d/P$. Thus, the state size of each Mamba-2 layer is $HPN = 2dN = 256d$. In transformer-based models, when using multi-headed attention, usually, the product of the head count $H$ and head dimension $P$ equals the hidden dimension $d$. Therefore, the KV cache of a transformer-based model is $2Td$, which means that when using the same hidden dimension, the state of a Mamba-2 layer is equal in size to a KV cache of 128 tokens.

Compared to many other recurrent models (e.g., the RWKV series (Peng et al., 2023a; 2024), GLA (Yang et al., 2023), and RetNet (Sun et al., 2023)), Mamba-2 does not have a state-less feed-forward network and has considerably more heads in each layer, making the state size much larger than other recurrent models. Compared to Mamba-1 (Gu and Dao, 2023), Mamba-1 uses $N = 16$, which means that the state size in Mamba-2 is 8 times larger than the state in a Mamba-1 model of roughly the model parameter count. Figure 12 shows the relationship between state size and model size of the RNN models in this study.

---

[7]https://tridao.me/blog/2024/mamba2-part1-model/

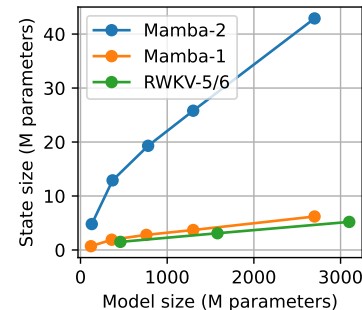

Figure 12: The relationship between state size and model size of various RNN models considered in this paper.

## A.2 SHORT CONVOLUTION

The $\text{Conv}(\cdot)$ function in Mamba-2 is a one-dimensional convolutional layer applied to each channel separately. For $i$-th channel, it can be formulated as follows.

$$y_{t,i} = \sum_{j=1}^{k} w_{j,i} x_{t-j,i} \in \mathbb{R}, \quad i = 1, \cdots, n_c$$

$k$ denotes the kernel size (set to 4 by default). $i$ denotes the channel index, $n_c$ denotes the number of channels. $y_{t,i} \in \mathbb{R}$ denotes the $i$-th channel of the output vector at $t$-th time step. $x_{t,i}$ represents the $i$-th channel of the input vector at $t$-th time step. $w_{j,i} \in \mathbb{R}$ denotes the $j$-th value in the convolutional kernel for channel $i$.

This model component accepts the last 4 token embeddings at the input. Therefore, it also has a state that contains information about the context, which we refer to as the *convolutional state*. To be concrete, due to information propagation through the layers, the short convolutional layer is a function of the last $4L$ tokens. For the 370M model size, this length is $4 \times 48 = 192$. Therefore, we can reasonably assume that this component contains much less contextual information relative to the recurrent state $h_t$. Thus, we have largely ignored this state in various discussions in this paper. However, we have also reported the distribution of the input to this short convolutional layer over time in Figure 17, for reference. As we can see, the convolutional state is relatively stable over time (compared to the recurrent state).

## B PASSKEY RETRIEVAL INFERENCE PARAMETERS

Throughout the whole paper, we use greedy decoding, not just for reproducibility, but also because our preliminary results show that other decoding parameters give noticeably worse performance on passkey retrieval.

We use 32-bit floating point precision for both model parameters and activations during inference, to ensure that precision errors do not introduce noise to the result. We have conducted some preliminary evaluations with BF16 and FP16 and found that there are no noticeable differences with using FP16, but computing some activations, especially the $\Delta_t$ and $\alpha_t$ with BF16 introduces an error around 1e-3. However, the explosion of channels in the states is consistently observed despite this precision error.

### B.1 PASSKEY RETRIEVAL PROMPT

The prompt that we use for the passkey retrieval task is as follows, using 34847 as the passkey for example, which is adapted from existing works (Zhang et al., 2024a). We also evaluate with slight variations to the template in preliminary experiments but do not observe considerable differences in the results.

```
There is important info hidden inside a lot of irrelevant text.
Find it and memorize it.
```

```
The grass is green. The sky is blue. The sun is yellow. Here we
go. There and back again.
...
The grass is green. The sky is blue. The sun is yellow. Here we
go. There and back again.
The passkey is 34847. Remember it. 34847 is the passkey.
The grass is green. The sky is blue. The sun is yellow. Here we
go. There and back again.
...
The grass is green. The sky is blue. The sun is yellow. Here we
go. There and back again.

What is the passkey? The passkey is
```

We sweep different context lengths $T \in \{1K, 2K, ..., 256K\}$, and for each length $T$, we generate $n$ prompts with evenly distributed needle positions, i.e., the $i$-th needle ($i \in \{0, \cdots, n-1\}$) of a sample is inserted at position $T \times i/n - 1$, from the beginning.

## C  MAMBA-2 WITH MODIFIED $\Delta_t$ ON PASSKEY RETRIEVAL

Ben-Kish et al. (2024) and GitHub user jzhang28[8] propose to improve Mamba's length generalization by reducing the value of $\Delta_t$. Ben-Kish et al. (2024) propose a heuristic method for identifying which head to modify and how to modify $\Delta_t$. However, their method requires task-dependent tweaking, so we do not consider comparing against it. jzhang28 propose to simply multiply $\Delta_t$ by a constant (they used 0.5). We apply this method and sweep different $\Delta_t$ for the best passkey retrieval performance, but they all result in worse performance than the original model across all context lengths.

## D  PASSKEY RETRIEVAL EVALUATION WITH OTHER ARCHITECTURES

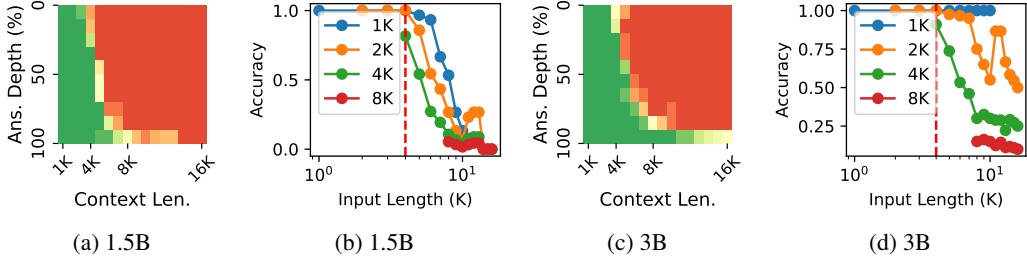

|  (a) 1.5B  |  (b) 1.5B  |  (c) 3B  |  (d) 3B  |

Figure 13: The performance of RWKV-5 official checkpoints on the passkey retrieval task. Each curve in (b) and (d) represents the accuracy of retrieving the needle when it is within the last $r$ tokens, with $r \in \{1K, 2K, 4K, 8K\}$.

Here, we also evaluate RWKV-5, RWKV-6, and Mamba-1 (some popular and strong RNNs) on the passkey retrieval task. The result is reported in Figure 13, 14 and 15. We can see that SE is observed in Mamba-1, but it is less severe for RWKV-5 and RWKV-6. We hypothesize that this difference is a result of architectural differences and that the state size is smaller in RWKV-5 and RWKV-6.

## E  PRE-TRAINED CHECKPOINTS

The pre-trained checkpoints used in our experiments are given in Table 1.

---

[8]https://www.github.com/jzhang38/LongMamba

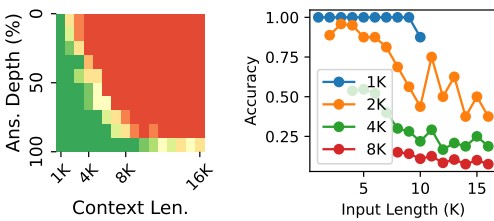

Figure 14: RWKV-6 1.6B result on the passkey retrieval task. The left plot shows the retrieval accuracy of the needle when it appears in the last $r = \{1K, 2K, 4K, 8K\}$ tokens.

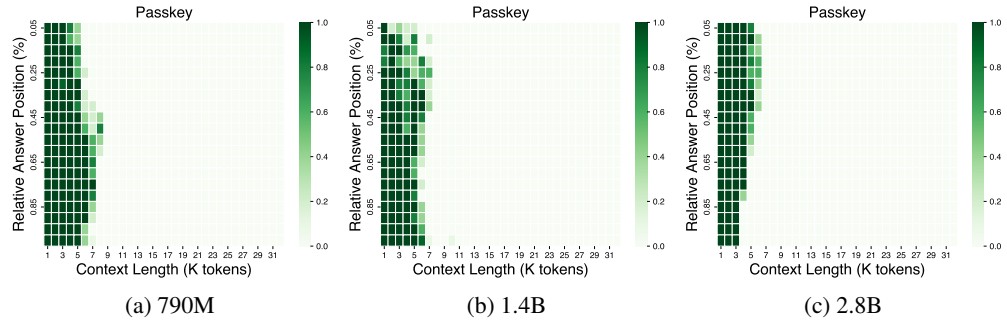

(a) 790M           (b) 1.4B           (c) 2.8B

Figure 15: The performance of Mamba-1 official checkpoints on the Passkey task. We can see a clear exhibition of state explosion, similar to Mamba-2.

| Model | Checkpoint URLs |
|---|---|
| RWKV-5 | https://huggingface.co/RWKV/rwkv-5-world-all-pth |
| RWKV-6 | https://huggingface.co/RWKV/v6-Finch-1B6-HF
https://huggingface.co/RWKV/v6-Finch-3B-HF |
| Mamba-1 | https://huggingface.co/state-spaces/mamba-130m
https://huggingface.co/state-spaces/mamba-370m
https://huggingface.co/state-spaces/mamba-790m
https://huggingface.co/state-spaces/mamba-1.4b
https://huggingface.co/state-spaces/mamba-2.8b |
| Mamba-2 | https://huggingface.co/state-spaces/mamba2-130m
https://huggingface.co/state-spaces/mamba2-370m
https://huggingface.co/state-spaces/mamba2-780m
https://huggingface.co/state-spaces/mamba2-1.3b
https://huggingface.co/state-spaces/mamba2-2.7b |

Table 1: The pre-trained checkpoints used in our experiments.

## F    EXPERIMENTAL DETAILS OF TRAINING ON LONGER SEQUENCES

We perform a hyperparameter search on learning rates, sweeping $\{1e-5, 2e-5, 5e-5, 1e-4, 2e-4, 5e-4, 1e-3\}$, selecting the best performing one by validation on passkey retrieval[9]. Regarding the WSD scheduler, it warms up linearly for 1000 steps and decays linearly with 50K steps. This setup is inspired by the authors of WSD (Hu et al., 2024).

Other hyperparameters are kept as similar to the original papers for Mamba-2 as possible. That means we use 0.5M tokens per batch because we found this to give more stable results for continual pre-training instead of the 1M batch size from the original paper. Training is done mainly in BF16,

---

[9]While the loss of many checkpoints was highly similar, their performance in passkey retrieval can differ a lot.

with some activations in FP32 (in the same manner as the official implementation). The optimizer is AdamW, with a 0.1 weight decay. Moreover, we use 1.0 gradient clipping.

All experiments are run on A800 80G, some are run with multiple nodes, and others with multiple GPUs on a single node.

### F.1 MODEL CONFIGURATIONS

For the models smaller than the 130M official checkpoint, we pre-train from scratch using the configurations reported in Table 2. We try to follow the same depth-to-width ratio found in the official checkpoints, although the ratio is not entirely consistent in those checkpoints. Hyperparameters not mentioned are kept the same as the 130M checkpoint.

| Model size | State size | # Layers | Hidden dim. | # heads |
|---|---|---|---|---|
| *Official checkpoints* | | | | |
| 780M | 19.3M | 48 | 1536 | 48 |
| 370M | 12.9M | 48 | 1024 | 32 |
| 130M | 4.8M | 24 | 768 | 24 |
| *Our checkpoints trained from scratch* | | | | |
| 84.6M | 2.4M | 12 | 768 | 24 |
| 47.0M | 1.6M | 12 | 512 | 16 |
| 36.4M | 0.8M | 6 | 512 | 16 |

Table 2: The configurations of the models used in finding the passkey retrieval memory capacity as a function of the state size.

## G STATE STATISTICS OVER CONTEXT LENGTH

Here, we provide a more detailed result on the inspection of SE over time.

Figure 16 shows the hidden state of the recurrent mechanism described in Eq. 3. Additionally, $B_t$, $C_t$, and $x_t$ in Mamba-2 are generated with a short channel-wise convolutional layer with a kernel size of 4:

$$B_t = \sigma(\text{Conv}[u_t W_B])$$
$$C_t = \sigma(\text{Conv}[u_t W_C])$$
$$x_t = \sigma(\text{Conv}[u_t W_x])$$

where $\sigma$ is the SiLU activation function. This function is also stateful because it operates on the last 4 tokens, therefore, we also collect the statistics of this convolutional state and report them in Figure 17. As we can see, the convolutional states are much more stable compared to the recurrent states. This is because only the last 4 tokens contribute to this state which avoids the explosion as a result of cumulative sum.

## H HGRN-2 LENGTH GENERALIZATION

We additionally evaluate HGRN-2(Qin et al., 2024) on the newlines prompt, and find that it also exhibits severe performance degradation on the "newlines" prompt. The model size is 1.3B. Perhaps surprisingly, the increase in perplexity happens considerably before the context length reaches the training length.

## I TRAINING-FREE LENGTH GENERALIZATION RESULT

Figure 19 reports the result of the training-free length generalization methods on Mamba-2 780M, evaluated on concatenated documents from RedPajama. We can see that while LongMamba[10] can

---
[10]https://github.com/jzhang38/LongMamba

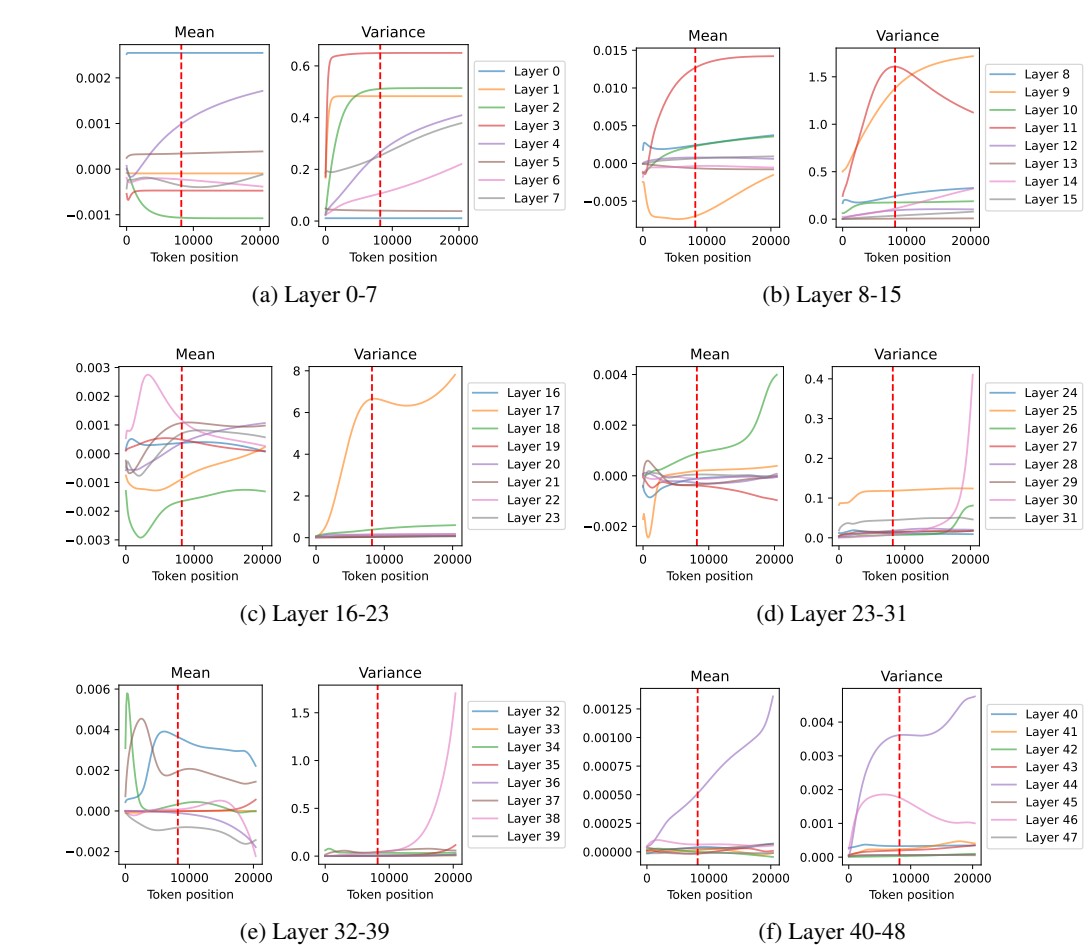

(a) Layer 0-7

(b) Layer 8-15

(c) Layer 16-23

(d) Layer 23-31

(e) Layer 32-39

(f) Layer 40-48

Figure 16: The mean and variance of the hidden state of each layer of Mamba-2 370M, computed on the "newlines" prompt.

greatly improve the length generalizability of the model by more than 3x, it causes noticeably greater perplexity on shorter sequences and still inevitably exhibits SE. All our methods successfully suppress SE, allowing the model to generalize to more than 64K tokens, although they underperform LongMamba on a range of context length (roughly from 8K to 20K).

## J THE "NEWLINES" PROMPT

In this paper, we collect the statistics of the state computed on a "newlines" prompt, a prompt where every token is the newline token, as shown below.

\n\n\n\n\n\n\n\n\n\n\n\n\n\n\n\n\n. . .

However, we again emphasize that SE is observed on prompts extracted from the pre-training corpus, the passkey retrieval task, or other randomly generated sequences. We have chosen the "newlines" prompt because the samples from the pre-training corpus are too short, and this prompt produces the most consistent and smooth layer statistics.

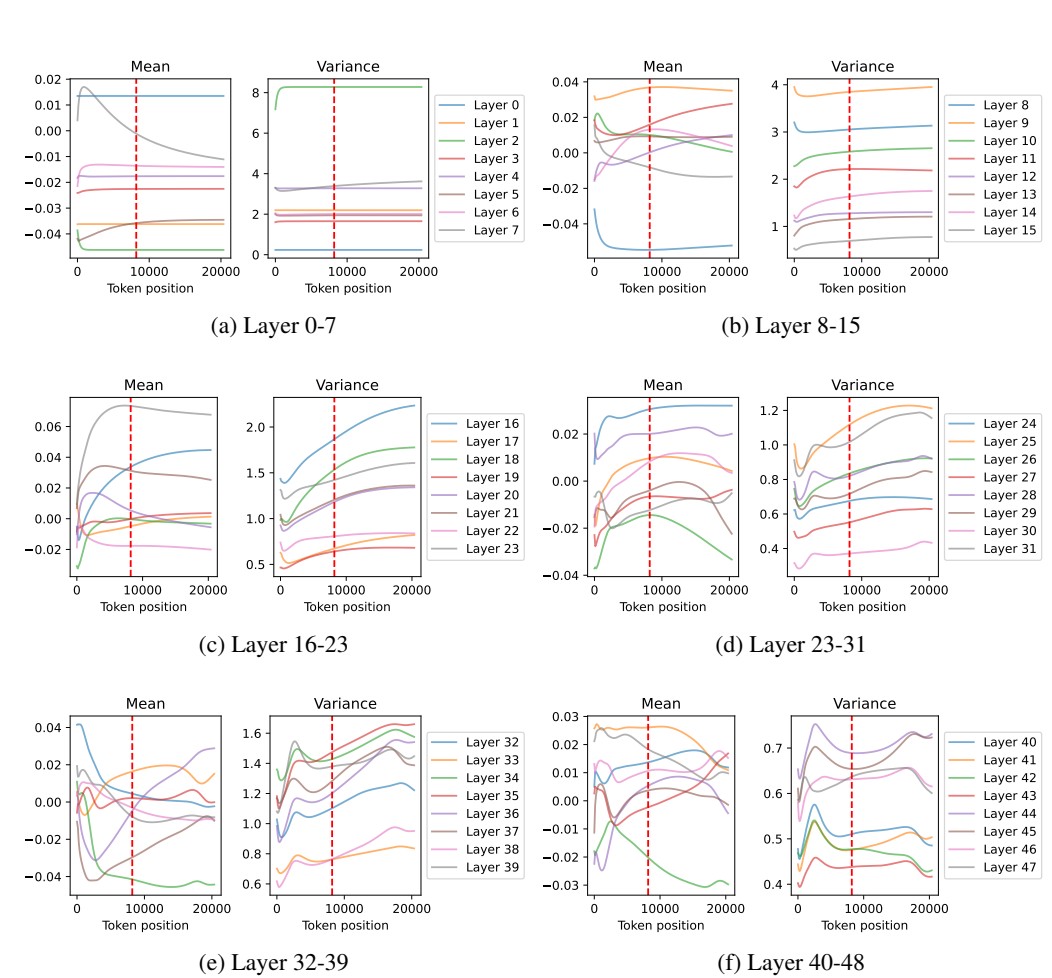

Figure 17: The mean and variance of the convolutional states (the representation of the last four tokens) of each layer in Mamba-2 370M, computed on the "newlines" prompt. We can see that the mean and variance are visibly more stable than the recurrent state.

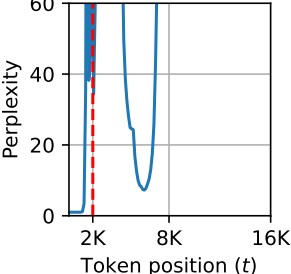

Figure 18: The perplexity of HGRN-2 1.3B on the "newlines" prompt as a function of time.

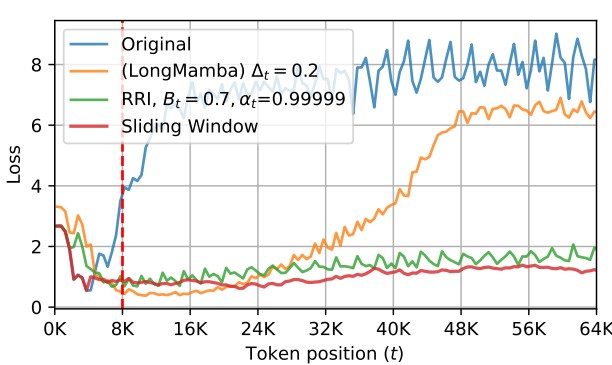

Figure 19: Result of training-free length generalization methods described in Section 4.4.2. The loss is computed on long documents from RedPajama (Computer, 2023). The red dotted line represents the training length.

