# OpenReview forum: "Stuffed Mamba: State Collapse and State Capacity of RNN-Based Long-Context Modeling"
_ICLR.cc/2025/Conference — Submitted to ICLR 2025_

### Official Review · Reviewer_NQ8J · 2024-10-24

**Soundness:** 3
**Presentation:** 3
**Contribution:** 3
**Rating:** 6
**Confidence:** 2

**Summary:**

This paper investigates a critical issue called "state collapse" (SC) in RNN-based language models, particularly focusing on Mamba-2. The authors identify that SC occurs when RNNs process sequences longer than their training length, causing severe performance degradation. They propose three training-free solutions and one training-based approach to mitigate SC, enabling models to handle sequences beyond 1M tokens. They conduct experiments to understand the relationship between state capacity and model size on the passkey retrieval task.

**Strengths:**

- The work addresses a crucial challenge in RNN-based language models that has not been solved. The authors clearly demonstrate the SC problem (Figure 1), showing Mamba-2 and RWKV suffer from performance degradation beyond their training length.
- The proposed solutions are simple and practical. The three training-free methods (adjusting memory retention, state normalization, and sliding window) are well-motivated by their analysis of the SC phenomenon. As shown in Figure 9, considering perplexity, these solutions effectively suppress SC and allow the model to generalize well in a long context.
- The empirical results on passkey retrieval are impressive. The authors achieved near-perfect accuracy on 256K context length with only a 370M parameter model (Figure 11).

**Weaknesses:**

- My major concern along this line of work, i.e., RNN-based language model, is their constant hidden state size. The fundamental limitation of constant state size in RNN-based models isn't fully addressed. When the task becomes complex and requires more memory, Mamba or RWKV cannot solve them well theoretically. However, attention may not suffer from this as their hidden state size is linearly large as the input token numbers. I am not sure whether RNN-based language model is a correct way. While the paper shows good results up to 1M tokens, there's no theoretical analysis of whether these models could scale to much longer contexts. The constant state size might ultimately prove to be a bottleneck that the proposed solutions can't overcome.
- The evaluation relies heavily on the passkey retrieval task and perpelxity, which is relatively simple compared to real-world long-context tasks. The paper doesn't demonstrate whether the proposed solutions work well for more complex tasks like long document summarization or multi-step reasoning. How would the proposed solutions perform on more complex long-context tasks? As the authors acknowledge in the limitations section, "high accuracy on this task may not reflect the capabilities of the model on real-world long-context tasks."
- The state capacity analysis in Section 5 lacks clear definitions and methodology. What is the rigorous definition of memory capacity in Figure 12? While Figure 12 shows relationships between state size and capacity, it's unclear how the authors precisely define and measure "memory capacity" or "state collapse." The paper would benefit from more rigorous definitions and evaluation metrics. Also, I would like to see some rigorous theoretical analysis, as the memory capacity seems can be fomularted as a well-defined information theory problem, e.g, theoretical upper bounds of the memory capacity for RNN-based models with constant state size.

**Questions:**

See Weaknesses above.

---

> ### Author Response · Authors · 2024-11-15
>
> Thank you for your thoughtful review. Regarding the mentioned weaknesses, we make the following response.
>
> **The fundamental limitation of constant-size memory.**
>
> The constant size of the recurrent state is indeed a fundamental bottleneck of RNN-based language models, and we are not claiming that it is the correct way ahead. This paper represents one of the first steps toward long-context modeling with recurrent language models. The constant-size memory is exactly the reason why we conducted the experiments described in Section 5, i.e., to estimate the training length at which the model can generalize and the context length at which the model can have near-perfect passkey retrieval accuracy (a very simple form of associative recall in natural language text). The result can provide important clues for choosing a suitable context length given the state’s constant size.
>
> Additionally, although the constant size of memory limits the capacity of RNN-based models, it also makes them much more efficient. Therefore, we believe the choice between RNNs and transformers is a trade-off between performance and efficiency, and perhaps some future architecture may be able to combine the best of both worlds. Our work is not meant to propose such architectures, but to analyze how and why RNNs exhibit length generalization failure.
>
> **Evaluation of passkey retrieval and perplexity may not reflect real-world tasks.**
>
> We have based our analysis on passkey retrieval and perplexity because we believe they are the minimum bar for memory recall. Failure in such tasks means that the model will likely fail on more realistic tasks. The main contribution of the paper is not to show that RNN-based language models are capable of handling realistic long-context problems, but that they are capable of performing the simplest kind of contextual information retrieval with lengthy contexts beyond 100K tokens. This is an ability that we have not observed in prior RNN-based models without specialized training.
>
> **Rigorous definition of state capacity**
>
> Thank you for bringing this up. The current draft of the paper has conflated two different notions of state capacity: the theoretical capacity of the state and the empirically estimated capacity of the state. In short, the "state capacity" mentioned in Figure 11, Section 5, and Section 7.2 all refer to the empirically estimated state capacity (by sweeping different training lengths). The measurement of state collapse is explained in Line 440. While a rigorous theoretical analysis is interesting, we have not included such an analysis in this work because our goal is to find the relationship between this estimated capacity and the state size. In Figure 11, we empirically find that this relationship is linear, which is a novel discovery. This relationship is an important consideration for RNN-based model training to avoid state collapse. An analysis of the theoretical state capacity does not change this conclusion, hence, we have chosen to leave that for future work.
>
> We will kindly notify you with another comment when we have updated the PDF file, and we will update all the reference to the paper in this response as well.
>
> Update: We have now revised the paper and updated the PDF file.

---

> > ### Comment · Reviewer_NQ8J · 2024-11-27
> >
> > Thanks for the authors' major revision. I agree with other reviewers that the modifications of the paper may need a thorough re-evaluation. I drop my confidence score from 3 to 2.

---

### Official Review · Reviewer_Eug9 · 2024-10-27

**Soundness:** 2
**Presentation:** 1
**Contribution:** 3
**Rating:** 3
**Confidence:** 4

**Summary:**

The paper looks into difficulties RNN-based language models have on long context tasks. First, the paper considers the observation that RNN based language models are unable to extrapolate to sequences longer than they were trained on. By studying the mean and variance statistics of activations in pretrained models, the authors identify an issue where some heads exhibit an exploding variance during extrapolation. Normalization then causes the information in the state to be erased. The authors argue this collapse is due to overparameterization and propose both training-free methods and a training related method to overcome this. The authors then attempt to characterize "state capacity" by sweeping training lengths and state sizes and observing the results of extrapolation. Empirical results suggest that several of the proposed methods can improve extrapolation ability.

**Strengths:**

- To my knowledge, this is one of the first works to take a systematic analysis of the model internals to determine the cause of RNN based models to fail to extrapolate
- The analysis of the mean and variance of the model activations is presented in a clear way making the cause of extrapolation failures clear
- The proposed training based and training-free interventions make sense and some of them appear to improve extrapolation based on the empirical results provided

**Weaknesses:**

- I would have liked to have seen a more precise definition of "state collapse" to provide clarity. Line 187 says it is: "a phenomenon that causes RNN models to exhibit abnormal behaviors on inputs longer than those seen during training." But what precisely constitutes an "abornomal behavior"?
    - Is state collapse simply the failure of the model to perform extrapolation, or is it a specific behavior of the model's state that leads to this failure? If the former, there may be no need for a new name, if the latter I would expect some characteristics of state collapse related to the state to be explicitly mentioned
    - Line 058 implies the extrapolation failure is "due to a phenomenon called state collapse", implying state collapse is different from the extrapolation failure itself. In addition, Section 4.2 implies that the cause of state collapse is the explosion of variance of some outlier heads. So based on these clues, we have the variance explosion causing "state collapse", and we have "state collapse" causing extrapolation failure. But this still leaves open the question of the precise definition of "state collapse"?

- In line 329 (Section 4.2.1), we are promised that in Section 4.3.2 we will be provided a proof (or perhaps an empirical observation, it is unclear) regarding an "if and only if" relationship between state size and state collapse. However this result is never provided in 4.3.2 (neither theoretical or empirical discussion), and no figures are referenced in this section. Then in Section 5, line 400 this "if and only if" result is referred to again as if it was described in Section 4.2.1. Was this accidentally left out? Or am I somehow missing it?

- I also found the use of "state capacity" to be imprecise. Line 074 implies that state capacity and state size are different (which makes sense). State capacity is referred to multiple times in Section 4 without definition. In line 401 we are told we can empirically estimate state capacity with sweeps. Finally, in lines 407-408 we are told that "we regard the minimum training length at which SC [state collapse] does not occur as the state capacity". So we have finally defined it in Section 5 by referring to "state collapse" which has also not precisely been defined as mentioned in the point above. In addition, the definition refers to an empirical measure, which we were previously told we would use to estimate "state capacity". It all seems very circular. It appears the paper is trying to distinguish between a theoretical state capacity and an empirical state capacity, however the two appear to be conflated.

-  There are many relevant prior works related to the long context failures of RNN based models that would have been good to reference to provide the reader with a clearer view of the literature in this area. Examples: https://arxiv.org/abs/2402.01032, https://arxiv.org/abs/2402.18510, https://arxiv.org/abs/2402.04248

- The paper only focuses on a notion of "state capacity" related to the ability to not spike perplexity on longer extrapolated sequences, or the ability to retrieve a single passkey across varying sequence lengths. This mostly seems to be related to the model's ability to ignore or forget irrelevant context. However another obvious version of "capacity" would be a multi-retrieval setting where the model has to use its fixed state to retrieve many passkeys or needles, even for shorter contexts (potentially within distribution, thus not required extrapolation), thereby stretching the memory capacity of the fixed state. This is explored synthetically in many works such as: https://arxiv.org/abs/2402.18668 for the multi-query associative recall task and in the RULER https://arxiv.org/abs/2404.06654 work which gives multi-key, multi-value and multi-query versions of passkey retrieval. This notion of capacity related to the actual information content that has to be remembered and recalled by the fixed state should at least be mentioned if not also ideally considered and analyzed in a definition of "state capacity".

**Questions:**

Most of my main questions and concerns I would like to have addressed are above in Weaknesses. Below is a summary of my main requests as well as a few additional questions.

1. Can you please provide a precise definition of "state collapse" as used in this paper?

2. Can you please clearly provide the proof or empirical observation regarding the relationship between state size and state collapse?

3. Can you please provide a precise definition of "state capacity" as used in this paper and ensure this is introduced earlier in the paper before it is used extensively? Can you also please clarify the relationship between state capacity, state size and state collapse as used in this paper? In addition, commenting on other notions of state/memory capacity as mentioned above would be helpful.

4. How is Method 1 of the training-free methods implemented in practice? Or what is the actual intervention? It is unclear from the text. I would recommend explicitly stating this in the Method 1 paragraph.

I like the general ideas of this paper and believe it can be interesting to the community. However I currently have concerns regarding the presentation of the ideas and technical terms in the paper as mentioned in the weaknesses section. This makes it confusing to read and clouds the message the authors are trying to get across. I can raise my score if these concerns are clarified and addressed.

---

> ### Author Response · Authors · 2024-11-15
>
> Thank you for the detailed and constructive review.
>
> **Regarding the precise definition of SC.**
>
> The precise definition of State Collapse (SC) is as follows:
>
> *SC refers to the exploding values in the recurrent state that causes performance degradation.*
>
> Moreover, with careful consideration, we have decided to rename SC to State Explosion (SE) in the updated paper, which we believe will make it clearer that the term refers to the value of the state and not the generalization failure. In the response below, we will use the new terms SE and SC interchangeably.
>
> In Section 4.2, when we say the "cause of SC", we refer to the cause of these exploding values, which is unexpected because the recurrent formula is stable as mentioned in Line 250. We acknowledge that the title of Section 4.2 is misleading, we are not claiming that the exploding variance is the cause of SE, this section is meant to illustrate the distribution of the state to better understand the phenomenon. In the updated paper, we will rename Section 4.2 to "Observation of State Explosion", and we will provide the above precise definition of SE in that section.
>
> **Explanation of Line 329 and Line 400.**
>
> The Section reference number is mistakenly incorrect, we are supposed to refer to Section 7.2. It is an empirical observation that we discover by training on different training lengths, and checking if SE is exhibited on the "newlines" prompt (as explained in Section 5). In Line 351, the claim about "if and only if" is an hypothesis, and is meant to be confirmed by the experiments in Section 7.2. We will rewrite the mentioned sections to make this clearer in the updated paper.
>
> **The Definition of State Capacity**
>
> We acknowledge that we have conflated two different concepts: the theoretical capacity of the recurrent state, and the empirically estimated state capacity. Thank you for bringing this up. In short, the "state capacity" mentioned in relation to state overparameterization, Line 408, Section 5, and Section 7.2 all refer to the empirically and indirectly estimated state capacity (by sweeping different training lengths). We will explicitly highlight this distinction in the updated paper (in Section 5).
>
> **Missing Related Works**
>
> Thank you for mentioning these papers. We will include all of them in the Related Works section in the updated paper and explain how they relate to and differ from our paper.
>
> **Other Notions of Capacity**
>
> Thank you for bringing this up. We want to first emphasize that the experiment regarding language modeling (Figure 11, left) is meant to show the empirical linear relationship between the training length threshold (beyond which SE is not exhibited) and the state size. This linearity between the training length threshold and the state size is a novel discovery, to the best of our knowledge.
>
> Regarding passkey retrieval, we chose to search for the empirical memory capacity on this task because we believe it can serve as the minimum standard for working associative recall. Namely, the failure in passkey retrieval will likely imply the failure in more challenging tasks such as multi-query retrieval and benchmarks such as RULER. We will dedicate a part of the updated paper to discussing the reason for choosing these tasks and how they are related to other tasks.
>
> **Questions 1 - 3**
>
> We believe these questions are addressed in the above comment. Thank you again for mentioning these points.
>
> **Question 4**
>
> For the SE mitigation method 1, the intervention is done by scaling $B_t$ and $\alpha_t$ with a multiplier. The multiplier is chosen by sweeping different values and validating on the "newlines" prompt. The final values are given in the legend in Figure 9.
>
> We will kindly notify you with another comment when we have updated the PDF file, and we will update all the references to the paper in this response as well.
>
> Update: We have now revised the paper and updated the PDF file.

---

> > ### Comment · Reviewer_Eug9 · 2024-11-25
> >
> > I thank the authors' for their response. I will take this into account when discussing with the other reviewers in the reviewer discussion phase. I believe this paper is improving a lot, however given how much it has needed to change from the original version, I suspect the paper would benefit from a resubmission at another conference.

---

### Official Review · Reviewer_hmho · 2024-10-28

**Soundness:** 2
**Presentation:** 2
**Contribution:** 3
**Rating:** 3
**Confidence:** 4

**Summary:**

This paper investigates the poor length generalization properties of recurrent models, in particular Mamba-2, and finds that a mechanism the authors term State Collapse (SC) is responsible. Based on this insight, the authors propose three mechanisms to mitigate state collapse in Mamba models, and find that it improves the length generalization capabilities of Mamba-2 models.

**Strengths:**

- The work tackles an important problem from a new direction, informing long-context training methods from mechanistic understanding of recurrent models.
- The newlines prompt is a compelling synthetic (if models can’t do this, something is wrong!) and provides focus and grounding to the work.
- The provided methods all appear to improve the length-generalization capabilities of Mamba-2 models. The authors provide analysis on both synthetics and a range of language modeling tasks.

**Weaknesses:**

- While the three proposed methods show promise, additional comparisons between them would enhance the paper's practical value. Figure 9 provides some insights, but more detailed ablation studies comparing their relative strengths and limitations would be valuable for practitioners.
- The generalizability of these methods beyond Mamba-2 presents an opportunity for future exploration. Given that RWKV and GLA are mentioned in the introduction, including experiments with these architectures could help demonstrate the broader applicability of the proposed approach.
- The methods currently show some performance trade-offs at shorter sequence lengths. Additional analysis or ablations of this phenomenon through the paper's theoretical framework could provide valuable insights and potentially suggest ways to maintain performance across all sequence lengths.
- The paper would benefit from engaging more with relevant prior work on recurrent neural network length generalization, including recent contributions such as Jelassi, Samy, et al. "Repeat after me: Transformers are better than state space models at copying." arXiv preprint arXiv:2402.01032 (2024); Arora, Simran, et al. "Zoology: Measuring and improving recall in efficient language models." arXiv preprint arXiv:2312.04927 (2023); and Ben-Kish, Assaf, et al. "DeciMamba: Exploring the Length Extrapolation Potential of Mamba." arXiv preprint arXiv:2406.14528 (2024). While this work offers a novel perspective, situating it within the broader literature would strengthen its contribution.
- The visualization of results, particularly in figures 2, 6, and 12, could be more accessible to readers. For instance, figure 2 would benefit from clearer labeling and explanation of metrics like "answer depth," and the color scheme implications could be more explicitly defined in the caption or text.

**Questions:**

- How general are these methods for recurrent models? Do they transfer well beyond Mamba-2?
- Of the proposed methods, which are more or less effective, and in which contexts?
- Is the tradeoff at short sequences inherent?

---

> ### Author Response · Authors · 2024-11-15
>
> Thank you for the thoughtful review.
>
> Before we respond to each of the mentioned weaknesses, we would like to emphasize the priorities of the paper because we believe that you have focused on parts that we believe are less important. The main contribution of the paper is:
>
> 1. The discovery and systematic analysis of state collapse (SC) (Section 4.1 and 4.2)
> 2. We present the state overparameterization hypothesis as an explanation for SC. This establishes a relationship between SC and the state capacity. (Section 4.3)
> 3. We empirically discover that there exists a minimum training length such that training lengths beyond this threshold result in a model without SC. Thus, this strengthens the hypothesis about state overparameterization (Section 7.2, Figure 9).
> 4. We empirically discover that this training length threshold is a linear function of the state size and that the model can recall contextual information beyond this training length (Section 7.2, Figure 11)
>
> The training-free mitigation methods (Section 4.4.2) are meant to showcase/strengthen the analysis conclusions we have made in Section 4.1 and 4.2, which is why we put it in a `\subsubsection`. In the updated version of the paper, we will revise the abstract and introduction to reduce the amount of content related to the training-free mitigating methods. We will move the evaluation result of these methods to the appendix because the paper will be more readable by using more space to explain our reasoning behind the state overparameterization hypothesis and empirical findings regarding the state capacity. Moreover, we will remove Method 2 (State Normalization) from the paper altogether, because the performance is bad and it is not the focus of our paper.
>
> In response to your summarization of weaknesses, we make the following remarks.
>
> **Comparison between the mitigation methods.** Due to limited space, we have not compared the performance of the mitigation methods, because they are less important, and the main take-away of the paper is the analysis of SC, the cause of it, and the training-based result.
>
> **Generalizability of mitigation methods.** Once again, we do not consider the generalizability of these methods because they are less important, and we have decided to spend more time and space on the analysis and the training-based experiment.
>
> **Trade-offs at shorter sequence lengths of mitigation methods.** This is an existing limitation, but since these methods are not the most important part of our paper, we leave such an exploration for future works.
>
> **Engaging more relevant prior works.** We will add the mentioned papers and discuss how they are relevant to our paper in the updated draft. Thank you for the suggestion.
>
> **Visualization of the results.** We will update the figures to make them clearer. The term “Ans. Depth” is used in existing works as well, so we thought it would be easily understandable. The color scheme will be added.
>
> **Questions**
>
> - **How general are these methods?** Regarding the training-free mitigation methods, as mentioned in Section 4.4.2, Method 3 is generalizable to all variants of RNN that can be written as a weighted sum of token representations, which include RWKV 4, 5, and 6, GLA, HGRN-2, RetNet, and many other contemporary RNNs. As for Method 1 and 2, it should be fairly easy to apply them to other RNN architectures, but that is outside the scope of this work.
> - **Of the proposed methods, which are more or less effective?** Our preliminary experiments show that Method 3 is the most performant in most cases, but we leave a more comprehensive comparison for future work.
> - **Is the tradeoff at short sequences inherent?** Method 1 and 2 will weaken the inserted information, therefore, it likely results in a performance drop across different context lengths. For method 3, the trade-off at shorter sequences can be entirely avoided by tweaking the sliding window size. We believe that the trade-off at short sequences is not inherent, but a more thorough exploration is outside the scope of our work.
>
> Update: We have now revised the paper and updated the PDF file.

---

### Official Review · Reviewer_gce7 · 2024-11-04

**Soundness:** 2
**Presentation:** 1
**Contribution:** 2
**Rating:** 3
**Confidence:** 4

**Summary:**

This paper studies the performance of language model architectures with constant state size (*e.g. Mamba).* It is well-known that performance of these architectures degrades on long sequence lengths. This paper focuses on this phenomenon, which the authors call “state collapse”. The paper makes three main claims:

1. That state collapse is due to the state size being *too big* during training, causing the model to
2. The problem can be mitigated post-training by (a) intervening on the strength of memory decay, (b) using state normalization, and (c) and converting the model to a sliding window model.
3. The problem can be mitigated during training by simply training on longer sequences.

**Strengths:**

**Significance.** The paper addresses a very important problem. The perplexity of a language model typically decreases as more context is given to the model (*i.e.* sequence length grows). However, this perplexity vs. token position plot usually flattens out or explodes for token positions beyond the sequence length on which the model was trained. A model with fixed recurrent state size for which perplexity *continues* to decrease indefinitely would represent a very significant advance.

- Note: This paper makes some steps towards realizing this objective, but as I mention in the weaknesses, lacks evidence that the problem is really solved.

**Interesting hypothesis for state collapse (Claim 1).** The authors argue that *“SC arises from state overparameterization relative to the training length. In other words, the state capacity is excessively large for the training length, allowing the model to achieve strong language modeling performance without learning how to forget when the state is about to overflow.”* This hypothesis for the cause of state collapse is interesting and, to my knowledge, original.

**Simple methods.** The mitigation methods proposed are simple and intuitive.

**Weaknesses:**

**Evidence for state over-parameterization (Claim 1) is unclear and limited.**

- *“Figure 7 shows the memory strength of the first token at different time steps, and we find that the exploded heads (heads 2, 4, and 7 in the 38th layer) have a strong inclination toward retaining all information within the training length, with a memory strength of over 0.8 at t=8K.”* It’s unclear how the layer/heads were chosen. Furthermore, the causal link between high memory strength in this one layer and state collapse is not evident.
- *“We also notice that Bt explodes earlier than ∆t. Therefore, we conclude that the collapse is largely attributable to Bt.”*  Looking at Figure 6 it’s not clear that $B_t$ “explodes” before $\Delta_t$. It also isn’t obvious why this would mean that the collapse is “largely attributable to Bt”.
- *“Finally, as we will show in Section 4.3.2, for any given training length, there exists a state size where SC will be exhibited if and only if the model’s state size is greater.”* Section 4.3.2 does not seem to address this point.
- *“It shows that SC is only exhibited by checkpoints beyond a certain amount of training, which coincides with behaviors of overfitting—a result of overparameterization.”* I found this argument confusing. Overfitting seems unrelated to the issue at hand. Training data in the language modeling setting is abundant, so it doesn’t seem like overfitting is part of the problem?

**Limitations with training-free mitigation methods (Claim 2).** There are significant limitations with the three training-free mitigation methods, which I am concerned may limit their utility in practice.

- **Methods 1 and 3 induce forgetting.** These methods work by explicitly inducing forgetting. This is concerning given that we don’t want the model to completely forget important earlier context (*e.g.* in passkey retrieval). **
- **Perplexity still rises on longer sequences with methods 1 and 3.** In Figure 9, it looks like theperplexity at high token positions (30k - 60k) is significantly higher than perplexity at earlier positions (10k). Ideally, the model’s perplexity would continue to decrease with longer sequences (or at least stay constant).
- **Method 2** **makes training/prefill non-parallelizable.** The authors did a good job of highlighting this limitation, but am raising it again here to emphasize how this would seriously complicate model training and serving.
- **Method 2** **degrades performance substantially.** Figure 9 shows that Method 2 leads to a significant increase in perplexity at token positions within the training length. This suggests that the model may need to adapt to state normalization *during training,* which is a concern given the non-parallelizability of the architecture.

**Evaluation on a limited set of tasks (Claims 2 + 3).** The experiments supporting *Claim 2* only measures perplexity on RedPajama. Given the comment above (that training-free mitigation methods cause forgetting) the methods should be evaluated on passkey retrieval, at a minimum. Ideally, the methods would also be evaluated on long context evaluations like ∞-BENCH or Loogle [[Zhang 2024](https://aclanthology.org/2024.acl-long.814.pdf), [Li 2024](https://arxiv.org/pdf/2311.04939)]. The experiments supporting *Claim 3* only measure perplexity on the synthetic *newlines* prompt*.*The lack of experiments on real data and hard long context tasks is concerning.

**Questions:**

- *Why are analyses performed at the 38th layer of the Mamba model?*
- *Can Figure 9 use a log scale on the y-axis?* It is hard to discern differences in perplexity.
- *What is the color scale in Figure 11 and which mitigation method was used for this experiment?*

---

> ### Author Response · Authors · 2024-11-15
>
> Dear reviewer, thank you for your detailed and thoughtful review. In response to your review, we would like to make the following remarks.
>
> **1 Evidence for state over-parameterization (Claim 1) is unclear and limited.**
>
> **It’s unclear how the layer/heads were chosen. Furthermore, the causal link between high memory strength in this one layer and state collapse is not evident.**
>
> The statistics of the state for every layer are already presented in Figure 17 in Appendix G. In short, several layers exhibited SC, and we arbitrarily chose one of the layers (layer 38) as an example due to limited space. The exact same analysis and findings can be made on other layers with SC as well (e.g., layer 9, 11, 17, 26, 30, etc). Similarly, we chose to only show the statistics on the first 8 heads to keep the plot simple, the same findings can also be made on other heads. Additionally, we emphasize that the correlation between high memory strength and SC is consistent in every layer. I.e., for any given layer, heads with SC have high memory strength and heads without SC have low memory strength.
>
> **Looking at Figure 6 it’s not clear that $B_t$ "explodes" $\Delta_t$ before. It also isn’t obvious why this would mean that the collapse is "largely attributable to $B_t$".**
>
> Thank you for this notice. Our claim about $B_t$ exploding before $\Delta_t$ is limited to the heads with SC. In Figure 6 (a), the curves correspond to the 8 heads shown in Figure 4, therefore, only the head 2, 4, and 7 (green, purple, and gray, respectively) are collapsing. Therefore, Figure 6 (a) shows that the explosion of $\Delta_t$ is only observed after 20K tokens, while in (b), we see that $B_t$ already shows considerable signs of explosion before 20K (the values are considerably larger than the one within the training length). However, we acknowledge that that claim about SC being largely attributable to $B_t$ is too bold and it will be removed in the updated paper.
>
> **Lack of support in Section 4.3.2**
>
> This is a mistake, this sentence is meant to refer to Section 7.2, in which we swept different training lengths, and empirically found that for each state size, there exists such a training length threshold beyond which SC is not exhibited.
>
> **Confusion about the state overparameterization argument. Overfitting seems unrelated to the issue at hand. Training data in the language modeling setting is abundant, so it doesn’t seem like overfitting is part of the problem?**
>
> The definition from Wikipedia is "An overfitted model is a mathematical model that contains more parameters than can be justified by the data."[1]. We are claiming that the state overfitted the training length because it contains too many parameters, resulting in an update rule (in the heads with SC) that fails to forget past information to avoid exploding values. The abundance of training data does not prevent this overfitting because, for the given training length (e.g., 8K), the distribution of the state seen during training is not varied enough to justify the state parameter count. This is a hypothesis we have made based on the inspection of the activations (Section 4.2), and then strengthened by Section 4.4, the mitigation methods, and the observation that there exists a training length threshold beyond which SC does not exhibit.
>
> **References:**
> [1] https://en.wikipedia.org/wiki/Overfitting

---

> ### Author Response · Authors · 2024-11-15
>
> (cont.)
>
> **2. Limitations with training-free mitigation methods (Claim 2).**
>
> In response to this weakness, we would like to first emphasize that the training-free mitigation methods are not the focus of this paper. They should only serve as an empirical demonstration of the insight that SE can be avoided by inducing forgetting (based on the analysis in Section 4.1 and 4.2). As a result, we have only dedicated one page of space to these methods. In practice, practitioners are encouraged to rely on training-based methods to mitigate SC (as described in Section 4.4). Training on longer sequences is not too expensive and can yield much better results. We kindly ask the review to focus on the experiments/findings related to SE and state capacity. To further emphasize the focus of this paper, we will revise the abstract and introduction to reduce the amount of content related to the training-free mitigation methods, and we will move the evaluation result of these methods to the appendix. We will also remove state normalization from the paper altogether because it performs poorly.
>
> - **Methods 1 and 3 induce forgetting.** Although these methods induce forgetting, they still outperform the base model with SC. This is a trade-off for the benefit of training-free nature. To mitigate SC without inducing forgetting, practitioners should rely on the training-based method.
> - **Perplexity still rises on longer sequences with methods 1 and 3.** This is an existing limitation of our method, however, this is already much better than the base model and the existing method.
> - **Method 2 makes training/prefill non-parallelizable.** Since the mitigation method is applied post-training, this only affects the prefilling phase, which is still a significant limitation. However, the same limitation applies to any non-linear RNN, and optimizing the efficiency of such architectures is an open problem that we do not tackle in this paper.
> - **Method 2 degrades performance substantially.** Yes, the model is not able to adapt to state normalization without training. However, with training, we believe it is better to simply use a much longer training length, as we did in Section 7.2.
>
> **3. Evaluation on a limited set of tasks (Claims 2 + 3).**
>
> Once again, the training-free mitigation methods are meant to strengthen/showcase our analysis conclusions, therefore, we did not evaluate them more thoroughly in the paper. We will evaluate these methods of passkey retrieval in the updated paper. In short, Method 1 and 2 have very poor retrieval performance, Method 3 can only correctly retrieve the passkey when it is located inside the window. For the training-based method, we will add an evaluation of RedPajama in the Appendix. Regarding other benchmarks, the models we experiment with are too small to have good results on more challenging tasks other than passkey retrieval.
>
> **Questions**
>
> - As stated, the analysis can be performed on any layer with heads whose state collapses. We arbitrarily chose one of the layers.
> - We have changed Figure 19 into the log-scale in the updated paper.
> - We have added the color scale in Figure 10. In brief, the value in each cell is very close to 1.0.
>
>
> We have now revised the paper and updated the PDF file.

---

> > ### Comment · Reviewer_gce7 · 2024-11-25
> >
> > Thanks to the authors for the thorough response.
> >
> > As I mentioned in my original review, the paper addresses a really important problem and it seems like the authors are making progress in tackling it.
> > However, it is still my feeling that the paper isn't ready for publication in this cycle. The original submission focused on the training free mitigation methods (which have significant limitations), while the resubmission focuses on the training-based mitigation method (which lacks thorough evaluation on more interesting tasks). The presentation of the paper could be improved in a resubmission by narrowing the scope to the training-based mitigation methods, and providing a more thorough evaluation of its performance.

---

### Official Review · Reviewer_xvuL · 2024-11-04

**Soundness:** 1
**Presentation:** 1
**Contribution:** 2
**Rating:** 3
**Confidence:** 2

**Summary:**

The paper investigates the length generalization problem in RNN-based models, specifically on the Mamba-2 series model. The authors identify a phenomenon they term state collapse, suggesting it as the primary cause of the model's failure to generalize to longer contexts. To address this issue, they propose four mitigation methods, including three training-free approaches. Experiments conducted on the RedPajama dataset indicate that with some of these methods, the Mamba-2 model of a specific size can improve its ability to generalize over longer sequences.

**Strengths:**

- The paper proposes to address an important yet underexplored issue of length generalization in RNN-based models.

- The authors identify a sharp change in the mean and variance of hidden states within Mamba-2 when processing long contexts. This observation provides a possible explanation of length generalization failures.

- Some proposed methods show positive results in mitigating length generalization issues in Mamba-2 on a specific dataset.

**Weaknesses:**

Although the problem is well-motivated, the presentation significantly hinders comprehension, making it challenging to follow the authors' reasoning and understand the experimental results. A primary issue is the lack of a formal, clear definition of the new phenomenon, State Collapse (SC), which should be introduced in the early sections. However, I found no clear definition, and the descriptions provided are vague, leaving readers to infer the meaning.

Moreover, I suspect that in the paper SC may be conflated with poor performance in length generalization. For example, in Section 4.2, the authors claim that "Figure 1 (a) and (b) shows SC on two different prompts". However, the figures appear to simply illustrate issues with length generalization. Due to this lack of clarity, it is difficult to fully assess the latter analysis of the potential causes of SC and the proposed methods to mitigate it. Even if I interpret SC as a sharp change in the hidden state values when processing long contexts, e.g., in Figure 4, 5, the latter analysis of SC in terms of over-parameterization and proposed solutions seem to be significantly disconnected from the phenomenon.

The proposed methods and experimental results are not entirely convincing, as they are only evaluated on a single model and dataset, limiting the generalizability of the findings. Additionally, the LongMamba method is confusing; it is not clearly described in the main text, leaving it unclear whether it serves as a baseline or represents an implementation of Section 4.3.2. When comparing with the baseline [1], the authors provide only a verbal discussion in Appendix C without any qualitative results to substantiate the claims, which makes it difficult for readers to evaluate the effectiveness of the proposed methods. Finally, regarding Method 1, the legend of Figure 9 includes hyperparameters that are not explained, making it unclear how sensitive the results are to these values. Clarifying these points would enhance the reader’s understanding and confidence in the results.

[1] Ben-Kish, Assaf, et al. "DeciMamba: Exploring the Length Extrapolation Potential of Mamba." arXiv preprint arXiv:2406.14528 (2024).

**Questions:**

Please see the weaknesses part.

---

> ### Author Response · Authors · 2024-11-15
>
> Thank you for your detailed review.
>
> **Definition of state collapse (SC)**
>
> Regarding the definition of State Collapse (SC), we acknowledge that we have mistakenly left out a precise definition in the paper. After a discussion between the authors, we have decided to make the term more precise by changing it to "State explosion" (SE), which emphasizes that the term describes an observed change in value. In the following response, we will use SE instead of SC. In the updated paper, we will add the following definition to Section 4: *SE is the phenomenon where the recurrent state exhibits exploding values, causing performance degradation.*
>
> **Difference between SE and length generalization failure**
>
> Thank you for raising this question. SE is one possible cause of length generalization failure, but it is not the same as length generalization failure. The reasons we hypothesize that SE is caused by overparameterization are the following empirical observations:
> 1. We observe that the performance drop is much more severe for models with larger state sizes.
> 2. We observe that during training, the performance within the training length consistently increases with the training amount, but the performance beyond the training length will first increase and then decrease after a certain amount of training. This implies that the model can overfit the training length, which is an indication of overparameterization. Since the contextual information is only stored within the recurrent state, it is reasonable to infer that the state is overparameterized.
> 3. The first token is not forgotten well beyond the training length, which indicates that the model is trying to retain all contextual information. This is only possible when the state is sufficiently large because, with a sufficiently small state (e.g., a 2x2 state), it will be impossible for the model to retrieve information with high precision from the state containing information about 8K tokens. Hence, we conclude that, without an overparameterized state, the model will learn to generate a sufficiently small memory decay value $\alpha_t$ (i.e., it induces more forgetting), which can help avoid the exploding values. The connection between smaller memory decay value and SE is highlighted in Figure 7, where the heads that exhibit SE have larger memory decay value (i.e., less forgetting). This strong correlation between larger memory decay values and exploded heads is consistently observed in different layers and model sizes.
> 4. Finally, and most importantly, we empirically observe that when sweeping different training lengths (Section 7.2), for every state size, there exists a training length threshold, beyond which the model does not exhibit SE.
>
> Therefore, when we claim that the state is overparameterized, we are claiming that the state is too large and is not sufficiently utilized for the given training length. This causes the model to generate memory decay values ($\alpha_t$) that are too large, which causes the state to explode due to the additive nature of the update rule.
>
> The updated version of the paper will change the content of Section 4.3 to highlight the above reasoning behind the overparameterization hypothesis.
>
> **Explanation of LongMamba and the reason to exclude another baseline.**
>
> LongMamba is a baseline and not an implementation of mitigation method 1. We do not consider [1] as a baseline because, as stated in Appendix C, it requires task-specific tuning, which is an entirely different application setting. Due to the limited space, we have kept the description of the mitigation method 1 short. In brief, method 1 simply scales $B_t$ and $\alpha_t$ by multiplying them with a scalar. We choose the two locally optimal sets of multipliers by validation on the newlines prompt. The legend in Figure 19 for Method 1 shows the different multipliers we arrived at. We will revise the paper to include this explanation.
>
> Finally, we would like to emphasize that the training-free mitigation methods are not the main focus of the paper, and kindly ask the reviewer to focus on the experiments/findings related to SE and state capacity when evaluating the paper. To make the priorities of the paper clearer, we have decided to revise the paper by moving the evaluation result of the training-free mitigation methods to the appendix, removing Method 2 (State Normalization) altogether, and using the extra space to provide an additional explanation of the findings in Section 5 and to explain the reasoning behind our state overparameterization hypothesis as mentioned above. We will also revise the abstract and introduction to mention less about the training-free mitigation methods
>
> We will kindly notify you with another comment when we have updated the PDF file, and we will update all the references to the paper in this response as well.
>
> Update: We have now revised the paper and updated the PDF file.

---

> > ### Comment · Reviewer_xvuL · 2024-11-26
> >
> > I thank the authors' efforts in clarifying the paper. As mentioned in my original review, the paper would greatly benefit from more careful organization and improved writing. With the paper's focus having shifted (e.g., from state collapse to state explosion, and the removal of training-free mitigation), I will retain my original score and suggest that the authors consider revising and resubmitting the paper to another venue.

---

### Author Response · Authors · 2024-11-16

Dear reviewers, we have revised the paper and updated the PDF file. For convenience of referencing, we hereby summarize the changes we have made to the paper:

- **Renaming State Collapse to State Explosion**: We have renamed “state collapse” to “state explosion” (SE) to better reflect the definition of the term. Also, we have added a precise definition of SE to the beginning of Section 4.
- **Training-free Mitigation Methods**: We have added more details about the training-free mitigation methods (explaining how the hyperparameters are chosen, and what they mean), and removed Method 2 completely (Section 4.4.2). Moreover, we moved the evaluation results of the training-free mitigation methods to the appendix (now Appendix I) to leave more space for other more important content.
- **Emphasis on the Focus of the Paper**: We have revised the abstract and introduction to reduce the amount of content related to the training-free SE mitigation methods to better reflect the focus of the paper (which is not the training-free SE mitigation methods). We have also revised the list of main contributions at the end of the introduction to emphasize the focus of this paper.
- **State Overparameterization Hypothesis**: Changed the previous Section 4.2.1 to Section 4.3 to highlight the importance of the state overparameterization hypothesis. We also added more explanation for the reasoning behind this hypothesis.
- **Various Renaming**: We have changed the title of Section 4.2 from “How is the Cause of State Collapse” to “Observation of State Explosion” to better reflect the purpose of the section. We also renamed Method 1 of the training-free methods to “Reduced Retention and Insertion Strength (RRI)” from “Forget More and Remember Less”, and Method 3 from “Sliding Window by State Difference” to “Sliding Window” (Section 4.4.2).
- **Critical Discovery of the Training Length Threshold**: We have added a clear description and emphasis on the critical discovery that there exists a training length threshold beyond which SE will not be exhibited. (Section 5)
- **Explanation of State Capacity**: We have added a definition to the state capacity mentioned in the paper (Section 5).
- **Figure Updates**: We added a colorbar to each of the passkey retrieval evaluation result figures. We also updated the perplexity evaluation figure of the training-free SE mitigation methods to improve readability (previously it was Figure 9, it is Figure 19 in the updated paper).
- **Explanation for the Choice of Layer and Head**: We added an explanation for why we chose the specific layers and heads in the main content. (Section 4.2)
- **HGRN-2**: We added an length generalization evaluation result of HGRN-2 in the appendix (Appendix H).

We will now revise all reference numbers (Section and Figure numbers) in our previous response to your reviews to match the updated paper.

---

### Meta-Review · Area_Chair_Ttv4 · 2024-12-09

**Metareview:**

The reviewers unanimously agree that the core idea presented in the paper is intriguing and has the potential to make a valuable contribution to the field. However, there is a shared concern that the paper appears rushed in its current form. This is evident in certain aspects of the presentation and writing style, which occasionally lack clarity and precision.

During the discussion phase, the authors demonstrated a commendable responsiveness to the reviewers' feedback by undertaking major updates to the manuscript. These revisions have notably improved the presentation of the paper.

Despite these positive changes, the reviewers maintain that the paper would benefit from a more thorough and meticulous rewriting process. This would ensure that the ideas are conveyed with the utmost clarity, rigor, and polish that they deserve. Therefore, the reviewers recommend that the authors resubmit the paper after undertaking a comprehensive revision and ensuring the overall presentation is of the highest quality.

**Additional Comments On Reviewer Discussion:**

Reviewers found the paper rushed and poorly written. The paper has undergone major changes during the discussion phase and reviewers felt the need for additional round of reviews.

---

### Decision · Program_Chairs · 2025-01-22

Reject